# Wind-driven snow conditions control the occurrence of contemporary marginal mountain permafrost in the Chic-Chocs Mountains, south-eastern Canada: a case study from Mont Jacques-Cartier

Gautier Davesne [1], [2], Daniel Fortier [1], [2], Florent Domine [2,3], James T. Gray [1]

[1] Cold Regions Geomorphology and Geotechnical Laboratory, Département de géographie, Université de Montréal, Montréal, Canada

[2] Centre for Northern Studies, Université Laval, Québec, Canada

[3] Takuvik Joint International Laboratory, Université Laval and Centre National de Recherche Scientifique, and Département de chimie, Université Laval, Québec, Canada

*Correspondence to*: gautier.davesne@umontreal.ca

**Abstract.** We present data on the distribution and thermo-physical properties of snow collected sporadically over 4 decades along with recent data of ground surface temperature from Mont Jacques-Cartier (1268 m a.s.l.), the highest summit in the Appalachians of south-eastern Canada. We demonstrate that the occurrence of contemporary permafrost is necessarily associated with a very thin and wind-packed winter snow cover which brings local azonal topo-climatic conditions on the dome-shaped summit. The aims of this study were (i) to understand the snow distribution pattern and snow thermo-physical properties on the Mont Jacques-Cartier summit; and (ii) to investigate the impact of snow on the spatial distribution of the ground surface temperature (GST) using temperature sensors deployed over the summit. Results showed that above the local treeline, the summit is characterized by a snow cover typically less than 30 cm thick which is explained by the strong westerly winds interacting with the local surface roughness created by the physiography and surficial geomorphology of the site. The snowpack structure is fairly similar to that observed on windy Arctic tundra with a top dense wind slab (300 to 450 kg m$^{-3}$) of high thermal conductivity which facilitates heat transfer between the ground surface and the atmosphere. The mean annual ground surface temperature (MAGST) below this thin and wind-packed snow cover was about − 1 °C in 2013 and 2014, for the higher, exposed, blockfield covered sector of the summit characterised by a sporadic herbaceous cover. In contrast, for the gentle slopes covered with stunted spruce (krummholz), and for the steep leeward slope to the southeast of the summit, the MAGST was around 3 °C in 2013 and 2014. The study concludes that the permafrost on Mont Jacques-Carter, most widely in the Chic-Chocs Mountains, and by extension in the southern highest summits of the Appalachians, is therefore likely limited to the barren wind-exposed surface of the summit where the low air temperature, the thin snowpack and the wind action bring local cold surface conditions favourable to permafrost development.

# 1 Introduction

The thermal impact of the seasonal snow cover is well-known as being one of the most critical factors for the spatial distribution of permafrost, especially in mountainous areas. Several studies have been undertaken on this topic in the European mountains (Haeberli, 1973; Grüber and Hoelzle, 2001; Luetschg et al., 2008; Farbrot et al., 2011; 2013; Hasler et al., 2011; Pogliotti, 2011; Gisnås et al., 2014; Ardelean et al., 2015; Magnin et al., 2016; Beniston et al., 2017), in Japan (Ishikawa, 2003; Ishikawa and Hirakawa, 2000), in the Canadian Rocky Mountains (Harris, 1981; Lewkowicz and Ednie, 2004; Lewkowicz et al., 2012; Bonnaventure et al., 2012; Hasler et al., 2015) and most recently in the Andes (Apaloo et al. 2012). The snow cover acts as a buffer layer controlling heat loss at the ground interface. It provides either a cooling (negative surface thermal offset) or warming (positive surface thermal offset) effect on the ground surface temperature (GST, see Table 1 for abbreviations used throughout this paper) whose magnitude depends on its depth, duration, timing and its thermo-physical and optical properties (Brown, 1979; Goodrich, 1982; Zhang, 2005; Ishikawa, 2003; Ling and Zhang, 2003; Smith and Riseborough, 2002, Hasler et al., 2011; Domine et al., 2015). All these snowpack characteristics – and the key parameters that control them, such as micro-relief, landforms, vegetation, and micro-climate – are strongly variable in space and time (Elder et al., 1991; Li and Pomeroy, 1997; Mott et al, 2010). The close link between snowpack thickness and ground thermal conditions was used to develop a map predicting the presence or absence of permafrost based on snow/ground interface temperature. The measurement of the *Bottom Temperature of Snow* (BTS) is the most widespread technique used to predict permafrost in mountain areas. It is well adapted to snowy environments, such as the Alps where it was developed, because a late-winter snowpack more than 80 to 100 cm thick is required to consider that the BTS values reflect the ground thermal condition and are decoupled from the atmosphere temperature (Haeberli, 1973; Hoelzle, 1992; Grüber and Hoelzle, 2001; Bonnaventure and Lewkowicz, 2008). In environments with thin snowpack, the BTS technique is not applicable, which led to the development of new techniques to predict permafrost occurrence such as the continuous monitoring of GST using temperature loggers (Hoelzle et al., 1999; Ishikawa and Hirakawa, 2000; Ishikawa, 2003; Gray et al., 2016).

In the Appalachian region of eastern North America, the existence of at least two bodies of contemporary permafrost are known with certainty: one of these occurs beneath the 1606 m high summit of Mount Washington in New England (Howe, 1971; Walegur and Nelson, 2003); the second beneath the 1268 m high summit of Mont Jacques Cartier, the highest point in south-eastern Canada situated in the Chic-Chocs Mountains (Fig. 1) (Gray and Brown, 1979, 1982). The latter site, with a well-documented record of geothermal data, is the location for the present study on the influence of the snow regime on marginal permafrost. Given its present ground temperature close to 0 °C (Gray et al., 2016), the occurrence and the spatial extent of this mountain permafrost body is thought to depend fundamentally on the existence of favourable azonal topo-climatic conditions on the dome-shaped summit. Similar exposed bedrock or mountain top-detritus summits where a permafrost body is marginally preserved have been reported in Scandinavia (e.g. Farbrot et al., 2011, 2013; Isaksen et al., 2011; Gisnås et al., 2013) and in Japan (Ishikawa and Hirakawa, 2000; Ishikawa and Sawagaki, 2001).

This study deals with the impact of the snowpack on the ground surface thermal regime and on the permafrost distribution on the Mont Jacques-Cartier summit, with implications on the potential presence of permafrost on the highest summits of the Appalachians. In order to address this question, it was necessary (i) to develop a qualitative and quantitative characterization of the snow distribution over the summit of Mont Jacques-Cartier and (ii) to quantify the surface thermal offset induced by the snow on the mean annual ground surface temperature (MAGST). The experimental design included snow thickness sounding, excavation of snow pits for observations of snow stratigraphy and measurement of density and temperature variations over the vertical profile, and GST monitoring based on the installation of temperature dataloggers. The hypothesis tested in this study is that wind driven almost snow-free conditions on the summit of Mont Jacques-Cartier explain the occurrence of this contemporary permafrost body due to intense ground heat loss in winter. Some studies have already mentioned the probable link between the near snow free winter conditions of the rounded summits and the occurrence of permafrost bodies in the Appalachian Range (Gray and Brown, 1979, 1982; Schmidlin, 1988; Walegur and Nelson, 2003), however the control induced by the snowpack and its feedback mechanisms on the winter GST over the summits of the Appalachians has not been quantified so far. The research reported in this paper is the first to have investigated this link in detail for one such summit dome, and to have quantified the multiple influences of the seasonal snowpack on the ground surface thermal regime. If our hypothesis can be confirmed, it will allow the prediction of favorable/unfavorable zones for permafrost occurrence over the entire dome-shaped summit of Mont Jacques-Cartier and on other high Appalachians summits from fine-scale spatial snow distribution patterns. New knowledge on the distribution of snow on the highest summits in South-Eastern North America is an important preliminary step in modelling the regional spatial distribution of permafrost, the evolution of the ground thermal regime and the future fate of mountain permafrost in the current climate change context.

**2 Physical characteristics of the Mont Jacques Cartier summit plateau**

The summit plateau of Mont Jacques-Cartier (1268 m a.s.l.) consists of an elongated, convex, low gradient surface 2.5 km long by 0.8 km wide, oriented NNE-SSW (Fig. 1). It is the highest of several domes rising above an extensive plateau surface, known as the McGerrigle Mountains. Geologically, the latter form a large, exposed, Devonian-age batholith, composed for the most part of granitoïd lithologies (De Romer, 1977), emplaced within the more extensive Chic-Chocs Mountains. The mean annual air temperature (MAAT) for the summit of Mont Jacques-Cartier for the period 1981-2010 was around − 3.3 °C according to the model presented by Gray et al., 2016. The calculation is based on the long-term air temperature time-series recorded by nearby coastal weather stations (Cap-Chat, 5 m a.s.l., and Cap-Madeleine, 29 m a.s.l.) and the local measured

surface environmental lapse rate (≈ 6 °C/km). This study focuses on GST recorded in 2013 and 2014 for which air temperatures were respectively 0.3 and 0.4 °C warmer than the reference period 1981-2010.

Because the current meteorological data are scarce in the Chic-Chocs Range, the amount and the distribution of precipitation is little known. The only regional study available to date has been made by Gagnon (1970) who concluded that the central mountains of Gaspésie likely receive annual precipitation in excess of 1600 mm. In winter, snowfalls are very frequent and abundant, as the whole Gaspésie region is in a corridor of low pressure systems developed by the contact between Arctic cold air masses from the north-west and Atlantic maritime cool air masses from the south-east (Hétu, 2007). Periods longer than one week without snowfall from December to March are rare. Some winter rainfall events occur each year, even on the highest summits, associated with brief periods of thaw under southerly flow conditions (Fortin and Hétu, 2009; Germain et al, 2009). The prevailing winds blow from the west and north-west according to the Limited Area version of the Canadian Global Environmental Multi-scale Model (GEM-LAM) for the period 2007-2010 (Bédard et al., 2013).

Above 1200 m a.s.l., the dome-shaped summit is characterized by a typical alpine tundra ecozone with various species of herbaceous plants (e.g. *Carex bigelowii*), mosses (e.g. *Polytrichium juniperinum*) and lichens (Payette and Boudreau, 1984; Fortin and Pilote, 2008). It is mantled by a cover of unconsolidated coarse angular clasts, forming blockfields (or *felsenmeer)* over about 30 % of the summit surface (Hotte, 2011; Charbonneau, 2015). In some areas, this sediment cover is re-worked by periglacial processes to form patterned ground features such as sorted polygons and block streams (Gray and Brown, 1979, 1982; French and Bjornson, 2008; Gray et al., 2009). On the margins of the dome-shaped summit, as the slope gradient increases, the lower part of the alpine belt presents a downward transition (between 1150 m to 1220 m *a.s.l.*) through isolated patches of stunted white and black spruce (*Picea glauca* and *Picea mariana*) to a continuous dense krummholz cover.

Despite the fact that isolated permafrost bodies are known or thought to exist on several summits in the northern Appalachian region in eastern North America (Brown, 1979; Péwé, 1983; Schmidlin, 1988; Walegur and Nelson, 2003), the Mont Jacques-Cartier summit is the only site equipped with a deep temperature cable that has been monitored continuously since 1977 (Gray and Brown, 1979, 1982; Gray et al. 2009, Gray et al., 2016). Early readings of the thermistors in the borehole drilled in 1977 (Gray and Brown, 1979, 1982) indicated a mean annual ground surface temperature (MAGST) around − 1.5 °C, an active layer thickness of ≈ 7 m and a Zero Annual Amplitude depth of ≈ 14 m (ZAA, as defined by van Everdingen, 1998). Extrapolation of the thermal gradient below ZAA suggested permafrost thickness possibly in excess of 45 m (Gray and Brown, 1979, 1982; Gray et al. 2009). With a ground temperature around − 0.3 °C recorded at 14 m depth in 2013, which is around 1 °C warmer than in 1977, the permafrost body of Mont Jacques-Cartier is presently degrading, displaying an overall warming trend over the last 37 years (Gray et al., 2016). If the warming trend continues over the next decades, the permafrost body can be expected to develop a supra-permafrost talik, become relict, and possibly thaw entirely in the near future (Davesne, 2015; Gray et al., 2016).

## 3 Methodology

In order to test our hypothesis proposing that the existence of contemporary permafrost on Mont Jacques-Cartier is dependent on favourable snow conditions, the link between the ground surface thermal regime and the snowpack characteristics has been explored. From this perspective, the following parameters have been measured or assessed for multiple locations across the plateau: (1) cumulative seasonal snow thickness; (2) the thermo-physical properties of the snowpack; (3) the seasonal timing and duration of the snowpack and (4) ground surface temperatures and their seasonal variability at sites presenting a wide range of snow depths.

### 3.1 Interannual snow thickness

This study relies on a compilation of data collected during fieldwork in the late winters of 1979, 1980, 2009, 2011, 2012 and 2014. Collecting data in winter on the summit, under extremely windy and cold conditions is difficult and only limited time could be devoted to that each day. The snow thickness on the Mont-Jacques Cartier summit was measured using a graduated probe 350 cm long. The surveys were generally conducted in late March or early April, when the snowpack reached its maximum depth. The snow sounding was made at regular intervals along several transects oriented WNW-ESE and NNE-SSW. Large-scale (1979, 1980, 2011 and 2012) and small-scale (1980, 2009, 2012) transects were made in addition to several random measurements. The snow thickness was also measured at each GST monitoring site in April 2014 (see section 3.3 below). Each measurement point has been geo-referenced using a global positioning system (GPS) and imported into a Geographic Information System (GIS, ArcGIS version 10.3.1). Furthermore, a Spot-5 satellite image taken on May 28th, 2013 (Google Earth, 2013) was used to identify the general snow ablation and accumulation areas on the summit. At that time of year, the areas where the snowpack was thin in winter were already snow-free, while those of preferential snow accumulations were still snow-covered.

### 3.2 Thermo-physical properties of the snowpack

A compilation of 9 snow profiles (6 in the alpine tundra zone and 3 in the krummholz), made during winter field studies in 1980, 2011, 2012, 2013 and 2014 on the Mont Jacques-Cartier summit and on the neighbouring summit of Petit Mont Saint-Anne (1147 m a.s.l.; Fig. 1), was used in order to analyse snow stratigraphy and to determine the physical properties of snow (density, grain size and snow crystal morphology, and temperature). Snow density $\rho_s$ (kg m$^{-3}$) was obtained in the field by weighing snow samples extracted from the layers of the snowpack using 2 cylinders of 227.1 and 742.5 cm$^3$, respectively. The grain size E (mm) and shape were determined in the field by placing a sample of snow on a millimetre-gridded plate and examining it with an 8x magnifying glass. The grain type identification and the graphical representation of each snow pit was achieved following the nomenclature given in the classification of Fierz et al. (2009). For indurated depth hoar, the graphical representation of Domine et al. (2016a and b) was used. Thermal properties were calculated using the afore-mentioned physical

properties. One of the most crucial variables conditioning the insulating capacity of the snowpack is its thermal resistance R ($m^2$ K $W^{-1}$) to heat flux transfers. The thermal resistance of a given snow layer $i$ is the ratio of its thickness $h_i$ over its thermal conductivity $\lambda_i$. For the whole snowpack, R is obtained by summing over all layers (Lunardini, 1981; Domine et al., 2012):

$$R = \sum_i \frac{h_i}{\lambda_i} \ , \tag{1}$$

To estimate the snow thermal conductivity, we used its correlation with density ($\rho_s$). Several equations relating both properties have been proposed, that of Sturm et al. (1997) being probably the most widely used. However, that equation is based on measurements containing a large proportion of snow samples from the taiga, where notoriously low thermal conductivity values are frequently encountered (Calonne et al., 2011, Domine et al., 2011). The equation of Domine et al. (2011) has been obtained from measurements of tundra snows, which are more similar to those we observed on Mont Jacques-Cartier. We therefore used their Eq. (2) and (3) here:

$$\lambda = 2.041*10^{-6}\rho_s{}^2 - 1.28*10^{-4}\rho_s + 0.032 \quad (30 \text{ kg m}^{-3} \leq \rho_s < 510 \text{ kg m}^{-3}) \ , \tag{2}$$

$$\lambda = 2.37*10^{-4}\rho_s + 0.0233 \quad (\rho_s < 30 \text{ kg m}^{-3}) \ , \tag{3}$$

The main physical variables that determine the metamorphic conditions of the snowpack are the temperature gradient in the snowpack and wind speed. Neither of these variables could be measured on Mont Jacques-Cartier and we have to rely on estimates to interpret the structure of the snowpack of the three snow pits presented in Figure 5. The average temperature gradient ($\Delta$T °C) through the snowpack was calculated using the Eq. (4):

$$\Delta T \, °C = (T_{ground \, surface} - T_{snow \, surface})/h \ , \tag{4}$$

where h is the snowpack thickness. The ground surface temperature data were used as an estimate for $T_{ground \, surface}$, air temperature as an estimate of $T_{snow \, surface}$. The onset of snow accumulation was estimated based on the results of the snowpack timing and duration analysis from 2008 to 2015. The maximum values of snow height were based on measurements made in April 2014. For the snow pit 1 site, the LT4 ground surface temperature data and a maximum snow height of 35 cm were used. We estimated that maximum snow height was reached in mid-December, after which wind erosion is assumed to have prevented any further accumulation. For the snow pit 2 site, the borehole data and a maximum snow height of 17 cm, reached at the end of November, were used. For the snow pit 3 at Petit Mt Ste-Anne, no datalogger was available at that site. We used instead ground surface temperature data from the LT1 datalogger installed on the southeast slope of Mont Jacques-Cartier, with similar snow height (maximum snow height of 200 cm, reached in mid-January) and vegetation.

## 3.3 Snowpack onset and melt analysis

The date of onset and complete melt of the seasonal snowpack was deduced from the daily GST recorded since 2008 by a datalogger (ACR Systems Smart Reader Plus 8) linked to a thermistor (Atkins type, accuracy of +/- 0.1 °C) installed near the ground surface at the summit of Mont Jacques-Cartier (Table 2). Generally, the date of the snowpack onset is detectable by

the smoothed profile of the daily fluctuations of the GST due to the thermal buffer created by snow (Teubner et al., 2015). Generally, a few centimeters of snow are sufficient to be detectable thermally. Furthermore, it is reasonably assumed that, as soon as the GST drops below 0 °C, precipitation falls as snow that persists on the ground unless an extensive period of positive air temperature follows. In spring, the date of the snowpack disappearance coincides with the time when the GST rises above 0 °C, generally after a brief zero curtain period.

## 3.4 Ground surface temperatures and their seasonal variabilities

The GST was recorded continuously from December 1$^{st}$ 2012 (3 readings per day) to 31$^{st}$ August 2015 by 20 dataloggers Trix-8 (LogTag®; resolution 0.1 °C, accuracy of +/- 0.5 °C) – named LT1 to LT20 – installed across the summit of Mont Jacques-Cartier (Table 2). In July 2014, an extra datalogger Trix-8 – named LT21 – was installed beneath a deep snow-bank on the southeast slope of Mont Jacques-Cartier. Each sensor was protected from humidity and ice by airtight plastic boxes, and all were installed about 5 cm below the ground surface to avoid any effect of direct solar radiation. The sensors were strategically located at sites with different surface characteristics (Table 2) representative of the summit's surficial geomorphology. Finally, a time-series of GST at the main borehole site on the summit is also provided by a thermistor cable linked to a datalogger (ACR Systems Smart Reader Plus 8) as mentioned above.

The GST time-series provided by the different dataloggers have been used to calculate the mean monthly GST, the mean winter (December-January-February [DJF]) GST (MGST$_w$) for the winter 2013-2014, the mean summer (June-July-August [JJA] GST (MGST$_s$) for the summer 2014 and the mean annual GST (MAGST) for the years 2013 and 2014.

To evaluate the importance of factors controlling the spatial evolution of the GST regime, a series of simple linear regressions was conducted between the MAGST in 2014, and 1) snow depth, 2) elevation and 3) potential incoming solar radiation (PISR). Elevation and PISR have been extracted from a 1:20 000 digital elevation model (DEM) in ArcGIS (version 10.3.1) using the module Spatial Analyst. Finally, the impact of the snow depth on the MAGST was assessed using the regression between snow depth measured in April 2014 and the MAGST of 2013 and 2014. A logarithmic regression was used as a best-fit model.

The main parameter describing the buffer effect of the snow cover on the ground temperature is the winter surface thermal offset ($\Delta T$, in °C), defined as the difference between GST and air temperature ($T_{air}$) (Eq. 5):

$$\Delta T = GST - Tair , \tag{5}$$

The air temperature dataset required to calculate the surface thermal offset was provided by a sheltered and ventilated temperature sensor installed 1 m above the ground surface near the summit of Mont Jacques-Cartier at 1260 m a.s.l. The air temperature has been recorded at hourly intervals since December 1$^{st}$ 2012 by a U22-001 (Hobo®; resolution of 0.2 °C, accuracy of +/- 0.21 °C).

The freezing n-factor ($nf$) was also used to evaluate the snow thermal effect on the ground. Its calculation is made from the sum of freezing degree-days at the ground surface (DDF$_s$) divided by the sum of the freezing degree-days of the air (DDF$_a$)

during the freezing season. The air and surface freezing season were considered to start when the mean daily air and surface temperature drops durably below 0 °C (at least 7 consecutive days) and to finish when the mean daily air and surface temperature becomes durably positive (at least 7 consecutive days) (Lewkowicz et al., 2012). Following this definition, we considered the air and surface freezing season separately. The ratio between $DDF_s$ and $DDF_a$ gives the *nf* as described by Eq. 6 (Karunaratne and Burn, 2003; Juliussen and Humlum, 2007):

$$nf = DDFs/DDFa , \tag{6}$$

Other parameters such as the winter *equilibrium temperature* (WEqT, in °C) and the timing and duration of the zero curtain effect have also been analysed. The WEqT reflects the thermal state of the underlying ground when the snowpack is thick enough to disconnect the GST from the air temperature. The zero curtain effect, which maintains the GST near 0 °C, is induced by the effect of latent heat release in fall when the active layer freezes and the latent heat consumption in spring during the snow melt (Hoelzle et al, 1992; Riseborough and Smith,1998; Ishikawa, 2003; Hasler et al., 2011).

## 4 Results

### 4.1 The snowpack distribution patterns

Visual observations, in-situ measurements and satellite imagery analysis clearly showed a variable pattern of snow accumulation over Mont Jacques-Cartier (Fig. 2 and Fig. 3). The thinnest snow cover was recorded on wind-exposed surfaces (slope < 15°) of the summit where an alpine tundra environment with blockfields and blockstreams is present (Fig. 4, A). The compilation of all snow measurements made over the dome-shaped summit gave an average snow thickness of 27 cm towards the end of winter, and well prior to the spring thaw (Fig. 4, B). As shown in the Spot-5 image from late May 2013 (Fig. 2), the wind-exposed summit is the first area to be snow-free in the spring.

Late winter values for snow thickness on the summit show low inter-annual variability in snowpack thickness (Standard deviation (SD) of 6 cm for the 5 different years of measurements) (Fig. 4, B). On a micro-scale however, snow probing showed considerable variations due to the surface roughness created by the large boulders associated with patches of felsenmeer. Field observations revealed that the spaces between blocks and linear depressions between blockstreams were filled with drifting snow early in winter whereas the top of the blocks and the linear crests of the blockstreams remained above the snow surface throughout the winter and were only covered with hoar frost and ice crusts (Fig. 3, A).

In the continuous krummholz zone, below 1150 m a.s.l., the mean snow thickness was ≈ 200 cm (Fig. 3, B; Fig. 4, A). In the discontinuous krummholz belt between 1150 to 1220 m a.s.l. – which marks the lower boundary of the alpine tundra environment – the isolated patches of shrubs induced localized thick (> 200 cm) snowdrifts (Fig. 3, B).

The highest snow accumulations occurred on the leeward southeast slopes of Mont Jacques-Cartier (Fig. 2). On the lower part of the concave southeast slope, mantled by blockfields and solifluction lobes, the snow cover was unusually thick, resulting in

a massive snow-bank which generally melted late in the summer (Fig. 3, C). For all the surveys made from 1978 to 2012, the maximum snow depth was greater than 350 cm (length of the snow probe).

## 4.2 Snowpack onset and melt

At the borehole site, on the wind-exposed summit of Mont Jacques-Cartier, the temperature data showed that the onset of the snowpack occurred at the end of October on average for the 2008-2015 period (Table 3). The earliest onset was during the winter 2009-2010 (11 October) while the latest was during the winter 2008-2009 (18 November) (Table 3). In spring, the snowpack melted out completely in mid-May on average over the 2008-2014 period (Table 3). The earliest complete snowpack melt out was recorded in spring 2013 (28 April) while the latest was in spring 2009 (25 May). On the most favorable snow accumulation areas - such as the headwall of a glacial cirque on the leeward south-eastern slope - the longest lasting snow patches are estimated regularly disappear by late summer according to quasi-annual observation between 1978 to 2015 by University of Montreal researchers.

## 4.3 Snow physical and thermal properties

On wind-exposed alpine tundra, the surface layers of the snowpack typically consisted of wind slabs comprised of small rounded grains (Fig. 5, A and B). A basal ice layer was often observed, and other ice layers or melt-freeze crusts could also be seen at several depths in the snowpack (Fig. 5, A and B). These may have been formed by rain on snow, freezing rain, supercooled fog events, or by radiative heating. The lack of complete meteorological data currently prevents the detailed understanding of the formation process of these layers. Indurated and poorly developed depth hoar, recognized as the metamorphism of a hard wind slab into depth hoar (Domine et al., 2016b), with crystal size rarely exceeding 2 mm, was often present near the base. Since indurated depth hoar is a snow type not mentioned in the international classification of Fierz et al. (2009) which focused on alpine snow, we represent it with the symbol proposed by Domine et al., 2016b), and which consists of a depth hoar symbol to which a dot representing a small rounded grain has been added. Higher up in the snowpack, progressively lower grades of crystal growth and facetization were observed, with faceted crystals 1 to 2 mm in size above the depth hoar, then faceted rounded crystal, and finally, on a frequent basis, hard wind slabs formed of very small sintered rounded grains. In the cavities between boulders, some large crystals (around 10-20 mm) of depth hoar were observed. In the alpine tundra zone the snow density at the end of winter was high, typically 350 kg m$^{-3}$ on average (SD = 80) according to measurements made in the 6 snow pits made between 1980 and 2014 (Fig. 5 and table 4). The thermal conductivity values ($\lambda$) calculated from the equation of Domine et al. (2011) derived from snow types fairly similar to those found here, range from 0.15 to 0.45 W m$^{-1}$ K$^{-1}$ with an average value of 0.28 W m$^{-1}$ K$^{-1}$ (SD =0.13). The values of the thermal resistance R for the snowpacks range from 0.45 to 2.45 m$^2$ K W$^{-1}$ with an average value of 1.67 m$^2$ K W$^{-1}$ (SD =1) (Fig. 5 and Table 4). As shown by Fig. 6 A and B, the calculated values of the thermal gradient fluctuated around 20 °C m$^{-1}$ for snow pit 1. Values were around

$40\ °C\ m^{-1}$ for snow pit 2 with occasionally high values up to $100\ °C\ m^{-1}$ through the winter 2013-2014. The first few days were removed from the analysis because small errors in the height of the thin snowpack can generate large errors in the temperature gradient.

On the krummholz and southeast leeward slope, the thick snowpack was composed of hard dense wind-packed snow layers and thin ice or melt-freeze layers (Fig. 5, C). In the lower part of the snowpack, crystal growth and facetization were moderate and led to crystals ranging from 2 mm depth hoar to 0.5 mm faceted rounded crystals. Further up, wind slabs comprised of small sintered rounded grains 0.2 mm in size were predominant, although slight faceting, occasionally leading to the formation of 1 mm faceted crystals, were also observed. Snow pits dug in 1980, 2011, 2012, 2013 and 2014 showed that the average density and thermal conductivity values of the snow layers were slightly higher than those measured in the alpine tundra area. However, given the much thicker snowpack, the thermal resistance R was significantly higher than in the alpine tundra area with values ranging from 4.5 to $18\ m^2\ K\ W^{-1}$ with an average value of $9.1\ m^2\ K\ W^{-1}$ (SD = 7.6) (Table 4). The calculated value of the thermal gradient for the snow pit 3 was around $10\ °C\ m^{-1}$, occasionally reaching and sometimes exceeding $20\ °C\ m^{-1}$, through the winter 2013-2014 (Fig. 6, C).

**4.4 Spatial change in ground surface thermal regime**

The spatial contrasts in the MAGST in 2013 and 2014 are illustrated in Fig. 7. The lowest MAGST was recorded over the wind-exposed summit while the highest values were recorded for the krummholz belt and the leeward slope of the summit where thick snow accumulates in winter. Similar MAGST spatial distributions patterns were observed in 2013 and 2014, suggesting low year to year variability (Fig. 7).

The MAGST spatial variability over the Mont Jacques-Cartier summit is therefore mainly explained by the high heterogeneity of the $MGST_w$. As illustrated by Fig. 8, in winter 2013/14, the GST distribution was highly spatially variable over the dome-shaped summit with a range of $\approx 14\ °C$ recorded between the coldest and the warmest sites (SD of 3.7). By contrast, in summer the GST was relatively homogenous spatially (SD of 0.8) (Fig. 8).

All the sensors installed on the wind-exposed surface of the summit (i.e. sensors LT2, 3, 4, 8, 10, 11, 12, 13, 14, 16, 17, 18, 19, 20), where the average snow thickness in April 2014 was 23 cm, recorded an average $MGST_w$ of $-13\ °C$ during winter 2013-2014 and $-11.8\ °C$ during winter 2014-2015 (Table A1). The $MGST_w$ was characterized by a low spatial variability over this alpine tundra zone. The winter surface thermal offset ($\Delta Tw$) was $+5\ °C$ for the winter 2013-2014 and $+5.8\ °C$ for the winter 2014-2015 on average. The freezing index $nf$ was 0.75 and 0.70 on average for the winter 2013-2014 and 2014-2015 respectively for the tundra zone which means that the snowpack provided a weak insulation to the ground surface. On an annual basis, the MAGST over the summit was around $-1.1\ °C$ on average in 2013 and $-0.9\ °C$ on average in 2014 with low spatial variability (Table A1).

Beneath krummolz patches (LT1) on the gentle slope around the summit and on the steep leeward slope (LT5, 6, 9 and 21), the $MGST_w$ was much higher than on the summit. For example, the sensor LT1 – where the snow depth was 260 cm in April
2014 – recorded an $MGST_w$ of $-1.2$ °C in winter 2013-2014, a $\Delta Tw$ of $+17$ °C and a *nf* of 0.08. At this site, the MAGST was 3.1 °C in 2013 and 2.9 °C in 2014 (Table A1). For the LT21, installed beneath the snow bank of the southeast slope, the $MGST_w$ was $-0.33$ °C in winter 2014-2015 with a *nf* of only 0.1. In these cases, the thick snowpack provided significant thermal insulation.

### 4.5. Evolution of the ground surface thermal regime through the freezing season

The sensors installed on the wind-exposed summit typically recorded rapid and short-term fluctuations of the winter GST following the air temperature evolution. The $\Delta T$ remained very low throughout the cold season and the GST values were accordingly very low (e.g. $GST_{mini}$ of $-30$ °C during the winter of 2013-2014 for LT13). The curves of daily cumulative $DDF_s$ and $DDF_a$ remained very close throughout the freezing season 2013-2014 (e.g. $DDF_a$ of 2650 and $DDF_s$ of 2231 for LT13; Fig. 9, A). Because of the thin snowpack, the winter equilibrium temperature (WEqT) was never reached, reflecting the limited
insulating effect of the snow cover.

Inversely, the sensors installed in areas which accumulated thick snow cover (e.g. LT1; LT21) exhibited a near stable GST close to 0 °C throughout the freezing seasons. The WEqT was reached as early as December because of the rapid build-up of the snowpack. Only the most pronounced air temperature variations had an impact on the GST but this was very gradual and time delayed. The positive $\Delta T$ was extremely high during the winter. The daily cumulative DDFs remained low through the
320 freezing seasons 2013-2014 for both sensors (e.g. $DDF_a$ of 2650 and $DDF_s$ of 210 for LT1) (Fig. 9, B). The duration of the zero curtain effect was 42 days in 2013 and 46 days in 2014.

### 5 Discussion

From field measurements coupled with the analysis of a satellite images, we deduced for the first time the pattern of snowpack distribution on the Mont Jacques-Cartier summit, and its linkage with measured GST. The results clearly show that the spatial
variability of the GST on Mont Jacques-Cartier is greatest in winter due to the heterogeneous distribution of the snow. The spatial distribution of the annual ground surface temperature and the mountain permafrost body is thought to depend fundamentally on the existence of favourable azonal topo-climatic conditions on the dome-like summit brought by the wind-driven, thin snow conditions. In this discussion, we will first examine the spatial distribution of the snowpack over the summit in relation to the main controlling parameters. In a second section, we will describe the metamorphism processes and the
specific physical properties of the snowpack on the site. In the third section, the thermal impact of the snowpack on the ground

surface temperature will be analysed and, finally, in the last section, we will propose a permafrost zonation for the Mont Jacques-Cartier based on the snowpack distribution.

## 5.1 Wind and the spatial distribution of snow

The prevalent west/north-westerly wind in conjunction with the summit topography and micro-relief are the major factors controlling the snow distribution patterns. Strong winds rapidly redistribute new snow. There is no anemometer on Mont Jacques-Cartier but wind data were obtained from the Limited Area version of the Canadian Global Environmental Multi-scale Model (GEM-LAM) with a 2.5 km horizontal uniform grid resolution for the period 2007-2010 (Bédard et al., 2013). This confirmed the windy character of the area, with hourly-averaged wind speeds > 8 m s$^{-1}$ 20% of the time and speeds > 10 m s$^{-1}$ 6% of the time. Such wind speeds are sufficient to erode most snows (Vionnet et al. 2012). Therefore, the wind-exposed surface of the Mont Jacques-Cartier summit is subjected to intense loss of snow by wind ablation, further exacerbated by snow sublimation. The measurements showed that the snowpack was quite homogenous with depths typically less than 30 cm at the end of the winter despite frequent and abundant snowfall in the Chic-Chocs Mountains (Gagnon, 1970; Hétu, 2007). The surface roughness associated with periglacial landforms – around 30-50 cm (Gray et al., 2016) – is the main factor controlling both the maximum thickness of the snowpack and its fine-scale variability. The measurements demonstrated that general snow patterns showed little variability throughout the winter and inter-annually (Fig. 4) because the controlling parameters, i.e. the wind action in conjunction with topography and surface roughness, do not change on a short-term basis.

In the preferential snow accumulation zones – i.e. in the krummholz patches and on the leeward convex-concave slope of the mountain – the snowpack attains thicknesses of > 200 cm. Because of its dense and tangled nature, the krummholz vegetation efficiently traps the blowing snow which forms hard wind slabs. Subsequently, the krummholtz shelters snow from wind erosion. Thus, krummholz distribution is a major factor explaining the snow accumulation patterns over Mont Jacques-Cartier. The height of the vegetation canopy, and the height at which the trunks are wind-blasted, tend to decrease at higher elevation in response to stronger winds and lower temperatures on the summit. On the northwest slope of Mont Jacques-Cartier, the krummholz typically attains heights of 2 to 3 m around 1100 m a.s.l. These observations are concordant with snow measurements in the krummholz belt. In the altitudinal transition between krummholz and alpine tundra, the krummholz cover becomes discontinuous leading to a heterogeneous snowpack, yet thicker than on the summit.

The snow swept away by winds from the west slope of the mountain and the dome-shaped summit is largely re-deposited on the leeward southeast slope of the Mont Jacques-Cartier. The convex-concave profile of the slope to the southeast of the summit creates a topographic depression filled by several meters of drifting snow throughout the winter. The resulting snow cover extends from the leeward side of the summit ridge to the upper alpine forest limit. This long-lasting snow patch generally melts late in the summer, but occasionally persists through the summer season (e.g. in 1977) following a particularly snowy

winter as reported by Gray and Brown (1982). Other major drifting snow deposits are present in the form of cornices on the high leeward edges of a small cirque on the southeast slope.

## 5.2 Metamorphism and physical properties of the snowpack

On wind-exposed alpine tundra on the summit of Mont Jacques-Cartier, the snowpack structure was similar to that observed on windy Arctic tundra, such as near Barrow, Alaska (Domine et al., 2012). Typically, Arctic snowpacks there are 15 to 40 cm thick and consist of a basal depth hoar layer of low density (200 to 300 kg m$^{-3}$) overlain by a denser (330 to 450 kg m$^{-3}$) wind slab comprised of small (0.2 mm) sintered rounded grains. Depth hoar forms when a high temperature gradient (typically > 20 °C m$^{-1}$ (Marbouty, 1980)) exists in a snow layer, and generates recrystallization of snow grains through sublimation and condensation processes (Sommerfeld and LaChapelle, 1970). Large faceted and striated crystals thus form as those observed in cavities between boulders on the summit of Mont Jacques-Cartier. This process also generates an upward water vapor flux that causes mass loss. The depth hoar can sometimes be indurated, *i.e.* it is formed when very high temperature gradients persist in a dense wind slab (Domine et al., 2016b). In that case, the depth hoar can have densities reaching 400 kg m$^{-3}$ and shows regions not affected by the upward water vapor flux. These regions still have small grains next to much larger depth hoar crystals (Domine et al., 2016b), giving the depth hoar a milky aspect. On the top of Mont Jacques-Cartier, we observed that the depth hoar was less developed and signs of melting were more frequent than on Arctic tundra. On the krummholz and southeast leeward slope, the thick snowpack had characteristics similar to those of alpine snowpacks, following the classification of Sturm et al. (1995), although densities were higher and snow layers harder.

Based on the estimated values of thermal gradient through the snowpack (Fig. 6, A), some assumptions are proposed regarding the metamorphic processes that occurred in the snowpack. For the snow pit 1 location (Fig. 6, A), the reconstructed thermal gradient values around 20 °C m$^{-1}$ are sufficient to form depth hoar in snows with $\rho_s < 350$ kg m$^{-3}$ (Marbouty, 1980), which likely explains the depth hoar observed at the base. For denser snows, higher gradients are required to form indurated depth hoar, although the threshold is not established. However, the observed facetization of the dense ($\rho_s > 350$ kg m$^{-3}$) upper snow layers are consistent with gradients > 20°C m$^{-1}$ persisting a large fraction of the time over long time periods. Very high temperature gradients (> 100 °C m$^{-1}$) are known to transform even melt-freeze crusts into depth hoar in a couple of weeks (Domine et al., 2009). The fact that such layers at the base of the snowpack remained recognizable at the end of winter confirms that such high gradients were not maintained for extended periods of time. The temperature gradient at the snow pit 2 location, near the borehole (Fig. 6, B), allowed the facetization of the very dense surface layers. Apart from a thin basal icy layer, no melt-freeze crust was observed, suggesting a transformation into faceted crystals by the strong gradients. In the thick snowpack of the krummholz belt (Fig. 6, C), the combination of moderate gradients with dense snow, as a result of snow compaction, prevented important depth hoar development, although occurrences of poorly developed depth hoar were observed.

## 5.3 Impact of the snowpack on the ground surface thermal regime

As expected, the snow depth was the main controlling factor of the MAGST with a $R^2$ of 0.81 (Fig. 10). The elevation is a secondary factor with a $R^2$ of 0.58 while the influence of potential incoming solar radiation (PISR) is minor with a $R^2$ of 0.09 (Fig. 10). The lesser importance of the elevation and especially of the incoming solar radiation could be explained by the fact that the study area is quite flat. The coefficients of correlation of these two parameters would have been higher in a context of complex topography. On the summit, the temperature gradient within the snowpack remained moderate. This reduces water vapor loss and therefore mass loss, as the temperature gradient generates a water vapor pressure gradient and therefore a flux resulting in mass loss. Important mass losses of basal snow layers are frequent in Arctic and subarctic regions (Domine et al., 2016b; Domine et al., 2015; Sturm and Benson, 1997) where low density layers of low thermal conductivity often develop at the base of the snowpack, creating an insulating layer that has the potential to limit ground cooling. We suggest here that the rapid ground cooling that prevented the establishment of durable elevated temperature gradients was facilitated by the peculiar morphology of the summit surface and by the strong winds. The felsenmeer landscape, with rocks protruding above the snow, created a lot of efficient thermal bridges that most likely greatly accelerated ground cooling (Ishikawa, 2003; Juliussen and Humlum, 2007; Grüber and Hoelzle, 2008; Gisnås et al., 2013). Furthermore, these rocks considerably increased surface roughness. Since turbulent heat fluxes (i.e. heat exchanges between the snow surface and the atmosphere) are proportional to wind speed and to surface roughness (Noilhan and Mahfouf, 1996; Vionnet et al., 2012), the summit morphology and the windy conditions are optimal to ensure rapid ground heat loss. In turn this heat loss limits the temperature gradient in the snow, slows down snow metamorphism and the formation of a low density layer at the base of the snowpack, so that there is a positive feedback between meteorological conditions and surface morphology on the one hand, and snow metamorphism on the other hand, which efficiently accelerates ground cooling and therefore promotes permafrost conditions. For these reasons, the snowpack buffer effect is very low and the ground surface thermal regime at the summit is strongly coupled with the atmospheric conditions (*e.g.* low surface thermal offset and high *nf*) involving very low $MGST_w$. In this zone, the winter equilibrium temperature (WEqT) is never reached (Fig. 9, A), which indicates sustained heat loss, and the zero curtain effect in spring is absent or very short (only a few days).

At lower elevations, where snow accumulates preferentially, the snowpack thermal resistance is higher than in the alpine tundra zone even though the snow is dense and hard and therefore has a high thermal conductivity. This is because the high snowpack thickness largely compensates its high thermal conductivity, and the resulting high thermal resistance is sufficient to prevent ground heat loss and ground freezing at depth. The ground surface thermal regime therefore shows a strong disconnection from the cold air temperatures, resulting in a strong positive surface thermal offset, a low *nf* and a $MGST_w$ close to 0 °C. In the snowiest zones, the WEqT was reached early in winter (e.g. in mid-November for LT1 in 2013, Fig. 9, B) and the zero curtain effect lasted more than 1 month (e.g. LT1 in spring 2013 and 2014, Fig. 9, B). As previously described by Ishikawa

(2003), a snow-ground temperature which remains close to 0 °C through the winter is clearly unfavourable to permafrost occurrence, especially in zones of discontinuous sporadic permafrost.

As shown in Fig. 11, a clear positive correlation exists between snow thickness measured in April 2014 over the Mont Jacques-Cartier summit and the MAGST in 2013 and 2014. According to the logarithmic curve, a snowpack exceeding about 40 cm induced a MAGST above 0 °C, which thus corresponds to the local snow depth threshold value for permafrost conditions. The high coefficient of determination for the equations ($R^2$ = 0.84 in 2013 and 0.77 in 2014) gives confidence in this threshold value. However, a note of caution is necessary, since only a few sensors (n=4) representative of the critical range of 40-50 cm snow thickness were available to build the plotted curve (Fig. 11). Furthermore, the precise snow thickness threshold for the occurrence of a negative MAGST is difficult to predict because the ground temperature evolves non-linearly with snow thickness. Other factors such as air temperature, the snow properties, the near-surface material properties and especially the snowpack timing and duration (Ling and Zhang, 2003) also have a great impact on MAGST. Smith and Riseborough (2002) estimated for example that a snow thickness of 25 cm is sufficient to prevent permafrost occurrence for a site with mineral soil and with a MAAT of − 2 °C similar to the conditions on Mont Jacques-Cartier. The higher value of critical snow thickness reported by the present study could be explained by the thermal effect played by the rock protruding above the snow and the turbulent heat fluxes as explained above.

The seasonal snowpack typically started late in October and melted in mid-May on the Mont Jacques-Cartier summit. The snow melt in spring is rapid; first because the snowpack is thin, and secondly, because the boulders protruding above the snow surface act as solar radiation absorbers and their roughness increases turbulent heat transfer from the warm atmosphere. Lateral heat transfer is also facilitated by the high thermal conductivity of granitic rock (Grünewald et al., 2010; Gray et al, 2016). The early and brief snow melt in spring is favourable to rapid surface warming because of the direct exposure to solar radiation, positive air temperature, the absence of zero curtain effect and the low amount of heat required to melt the thin snowpack (Ling and Zhang, 2003). However, the warming effect of the ground surface induced by the early melt of the thin snowpack on the wind-exposed surface is not sufficient to counteract the intense cooling effect which takes place throughout the winter.

## 5.4 Permafrost zonation based on the snowpack distribution

A MAGST below 0 °C is favourable to the occurrence of discontinuous mountain permafrost (Abramov et al., 2008). For the years 2013 and 2014, the sensors that monitored a MAGST below 0 °C were in the thin snow zones of the Mont Jacques-Cartier summit. As shown by Fig. 12, the snow distribution is clearly a reliable indicator for mapping the predicted MAGST and probable permafrost distribution over the Mont Jacques-Cartier and other surrounding high summits of the Chic-Choc Range and of the Appalachians. As the thickness of the snow cover depends on physiography (wind-exposed and sheltered zones), and micro-topography (e.g. bouldery felsenmeers or smooth bedrock surfaces, alpine tundra or krummholz vegetation

cover), the variability in these terrain factors around the summit dome plays an important role in circumscribing the extent of the permafrost body.

Based on the satellite image Spot-5 taken on May 28[th] 2013, the potential extent of permafrost over Mont Jacques-Cartier
summit can be inferred from the higher limit of the krummholz belt and the distribution of the snowpack at the end of the spring 2013 (Fig. 12). At that time of the year, only the less snowy zones, i.e. the wind-exposed dome-shaped summit, are already snow free. These zones are the most favourable to permafrost preservation as suggested by the MAGST recorded below 0 °C in 2013 and 2014. Inversely, the zones with remaining snow had thicker snowpacks and their MAGST are too high to allow permafrost preservation (Fig. 12). According to Fig. 12, the potential permafrost body extends over 1.5 km$^2$ on Mont
Jacques-Cartier which is slightly lower than the previous estimation of 1.8 km$^2$ made by Gray et al. (2009).

Due to the extremely thin snow cover and the relatively constant inter-annual pattern of late winter snow depths measured over the barren summit of Mont Jacques-Cartier, we suggest that the evolution of this mountain permafrost body is closely coupled with the trends in air temperature at the site. Furthermore, because the granitic bedrock which composes Mont Jacques-Cartier has a high thermal conductivity (2 to 2.7 W m$^{-1}$ K$^{-1}$) and a low ice content (Gray and Brown, 1979), the response of the thermal
regime of the permafrost to fluctuations of the air temperatures is extremely rapid. Therefore, the marginal permafrost body of Mont Jacques-Cartier becomes an excellent indicator of regional climate change, as suggested by Gray et al. (2016), who demonstrated its recent rapid warming over the last decade following the air temperature trend.

**6 Conclusion**

This study represents the first analysis of the impact of snow conditions on the ground thermal regime and permafrost over a
470 rounded summit of the Appalachian Range. Overall, the results showed that the snow distribution pattern across the summit of Mont Jacques-Cartier controls the small-scale spatial variability of the MAGST. This pattern is therefore of paramount importance in estimating the spatial limits of this permafrost body, the southernmost in eastern Canada. The snow thickness on the summit is dependent on wind action in conjunction with the local topography, surface roughness and vegetation. Because these controlling conditions are quite stable over time, the general pattern of snow distribution tends to repeat itself
475 year after year over the summit. On the wind-exposed surface of the summit, the thin, highly thermally conductive ($\lambda = 0.28$ W m$^{-1}$ K$^{-1}$) and discontinuous snowpack leads to a strong connection between winter air and ground surface temperatures. The thin snow cover favours intense ground heat loss, low MAGST$_w$ ($\approx$ 15 °C colder than areas with thick snowpack) and deep frost penetration in winter. This is exacerbated by thermal bridging caused by the protruding boulders of the felsenmeer, sorted polygons and blockstreams. Furthermore, these rocks increase surface roughness and therefore turbulent heat exchange with
480 the atmosphere, accelerating ground cooling in winter and snow melt in the spring. We also propose that the cooling effect of blocks, by reducing the temperature gradient in the snowpack, modify snow metamorphism and significantly reduce the

formation of an insulating basal depth hoar layer, so that interactions between snow and the rough rocky surface combine to create a positive feedback effect that optimize ground cooling. In the krummholz belt around the summit and on the leeward slope of the mountain, the snow drift accumulations are in excess of 200 cm thick. Such a snowpack has a thermal resistance 5 times as high on average as in the alpine tundra zone, inducing strong positive surface thermal offset on the GST ($nf$ close to 0) and considerably reducing ground heat losses during the cold season. The permafrost body on Mont Jacques-Cartier is therefore very likely limited to the barren wind-exposed surface of the summit where the snow thickness is lower than 30-40 cm. Due to the limited thermal buffer played by the thin snowpack, the thermal regime of this permafrost is quasi directly connected to the air temperature. The permafrost on Mont Jacques-Cartier is thus very sensitive to recent climate warming and its degradation could lead to major changes in soil hydrology, geomorphology and alpine geosystem dynamics. In this context, this study provides the first step in the future regional mapping of discontinuous sporadic permafrost in Eastern North America which would identity alpines environments susceptible to be affected by permafrost degradation.

**Acknowledgements**

We thank the Parc National de la Gaspésie, especially, Francois Boulanger, Pascal Lévesque and Claude Isabel for their assistance with logistical aspects of the research and for the use of facilities. This project was carried out with the support of the Natural Science and Engineering Research Council of Canada (NSRC) to D. Fortier and of the Faculté des études supérieures et postdoctorales of the Université de Montréal to G. Davesne. We also thank all graduate and undergraduate students of the Université de Montréal who participated to field campaigns from 1978 to 2014. Finally, we gratefully acknowledge the helpful comments of two anonymous reviewers and of the Editor, K. Isaksen.

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

**Table 1. Abbreviations used in this paper.**

| Abbrev. | Definition |
|---------|------------|
| DDFs | sum of freezing degree-days at the ground surface |
| DDFa | sum of freezing degree-days of the air |
| DDTs | sum of thawing degree-days at the ground surface |
| DDTa | sum of freezing degree-days of the air |
| GST | ground surface temperature |
| MAAT | mean annual air temperature |
| MAGST | mean annual ground surface temperature |
| MGSTw | mean winter ground surface temperature |
| MGSTs | mean summer ground surface temperature |
| nf | Freezing n-factor |
| SD | Standard deviation |
| WeqT | Winter equilibrium temperature |

**Table 2. Detailed information of sensors location**

| ID | Record period [dd/mm/yy] | Elev.(m) | Aspect | Slope (°) | Vege. Type | Ground surf. |
|----|--------------------------|----------|--------|-----------|------------|--------------|
| ACR | 01/09/08 to 31/08/15 | 1268 | E | 2.5 | - | blockfield |
| LT 1 | 01/12/12 to 31/08/15 | 1196 | NW | 16 | krummholz | organic |
| LT 2 | 01/12/12 to 31/08/15 | 1222 | NW | 19 | herbaceous | blockfield |
| LT 3 | 01/12/12 to 31/08/15 | 1252 | N | 11 | - | blockfield |
| LT 4 | 01/12/12 to 31/08/15 | 1258 | SE | 9 | - | blockfield |
| LT 5 | 01/12/12 to 31/08/15 | 1243 | SE | 12 | - | blockfield |
| LT 6 | 01/12/12 to 31/08/15 | 1219 | SE | 23 | sparse shrubs | blockfield |
| LT 8 | 01/12/12 to 31/08/15 | 1265 | SE | 4 | herbaceous | blockfield |
| LT 9 | 01/12/12 to 31/08/15 | 1226 | SE | 21 | - | blockfield |
| LT 10 | 01/12/12 to 31/08/15 | 1257 | S | 11 | - | blockfield |
| LT 11 | 01/12/12 to 31/08/15 | 1255 | SW | 9 | - | blockfield |
| LT 12 | 01/12/12 to 31/08/15 | 1261 | SW | 7 | - | blockfield |
| LT 13 | 01/12/12 to 31/08/15 | 1265 | W | 2 | herbaceous | blockstream |
| LT 14 | 01/12/12 to 31/08/15 | 1265 | W | 2 | - | blockfield |
| LT 15 | 01/12/12 to 30/11/13 | 1264 | N | 4 | - | blockfield |
| LT 16 | 01/12/12 to 31/08/15 | 1248 | NE | 4.5 | herbaceous | sorted polygon |
| LT 17 | 01/12/12 to 31/08/15 | 1228 | NW | 9 | - | blockstream |
| LT 18 | 01/12/12 to 31/08/15 | 1234 | N | 1 | herbaceous | blockstream |
| LT 19 | 01/12/12 to 31/08/15 | 1237 | SW | 1 | herbaceous | sorted polygon |
| LT 20 | 01/12/12 to 31/08/15 | 1236 | E | 3 | herbaceous | sorted polygon |
| LT 21 | 01/09/09 to 01/09/10 | 1185 | E | 21 | sparse shrubs | blockfield |

**Table 3.** Timing and duration of the seasonal snowpack at the borehole site derived from the ground surface temperature recorded over the 2008-2014 period.

| Winter | Onset [dd-mm] | Melt [dd-mm] | Duration [day] |
|---|---|---|---|
| 2008/2009 | 18-Nov | 25-May | 188 |
| 2009/2010 | 11-Oct | 22-May | 223 |
| 2010/2011 | 20-Oct | 17-May | 209 |
| 2011/2012 | 26-Oct | 7-May | 193 |
| 2012/2013 | 13-Nov | 28-Apr | 166 |
| 2013/2014 | 2-Nov | 10-May | 189 |
| 2014/2015 | 23-Oct | 9-May | 198 |
| *Mean* | *29-Oct* | *12-May* | *195* |
| *Min* | *11-Oct* | *28-Apr* | *166* |
| *Max* | *18-Nov* | *25-May* | *223* |

**Table 4.** Details of the snow physical and thermal properties measured and calculated for the 9 snow pits made on Mont Jacques-Cartier (MJC) and Petit Mont Saint-Anne (PMSA). The snow density ($\rho_s$) and the thermal conductivity ($\lambda$) of each snowpack layers have been averaged for each snowpack. R represents the thermal resistance of the snowpack.

| Date | Location | Thickness [m] | $\rho_s$ average [kg m$^{-3}$] | $\lambda$ average [W m$^{-1}$ K$^{-1}$] | R [m$^2$ K W$^{-1}$] |
|---|---|---|---|---|---|
| March 1980 | MJC - alpine tundra | 0.24 | 285.42 | 0.16 | 2.45 |
| March 2010 | PMSA - alpine tundra | 0.48 | 430.19 | 0.44 | 1.95 |
| March 2011 | MJC - alpine tundra | 0.11 | 331.43 | 0.26 | 0.45 |
| March 2012 | MJC - alpine tundra | 0.42 | 272.62 | 0.15 | 2.9 |
| April 2014 | MJC - alpine tundra | 0.38 | 336.29 | 0.2 | 1.73 |
| April 2014 | MJC - alpine tundra | 0.17 | 473.54 | 0.44 | 0.51 |
| | *Mean* | *0.30* | *354.92* | *0.28* | *1.67* |
| | *SD* | *0.15* | *80.30* | *0.13* | *1.00* |

| Date | Location | Thickness [m] | $\rho_s$ average [kg m$^{-3}$] | $\lambda$ average [W m$^{-1}$ K$^{-1}$] | R [m$^2$ K W$^{-1}$] |
|---|---|---|---|---|---|
| March 1980 | MJC - SE slope | 3.45 | 340 | 0.23 | 17.86 |
| March 2010 | PMSA - krummhloz | 1.23 | 383.53 | 0.29 | 4.51 |
| April 2014 | PMSA - krummholz | 1.73 | 431 | 0.36 | 4.93 |
| | *Mean* | *2.14* | *384.84* | *0.29* | *9.10* |
| | *SD* | *1.16* | *45.51* | *0.07* | *7.59* |

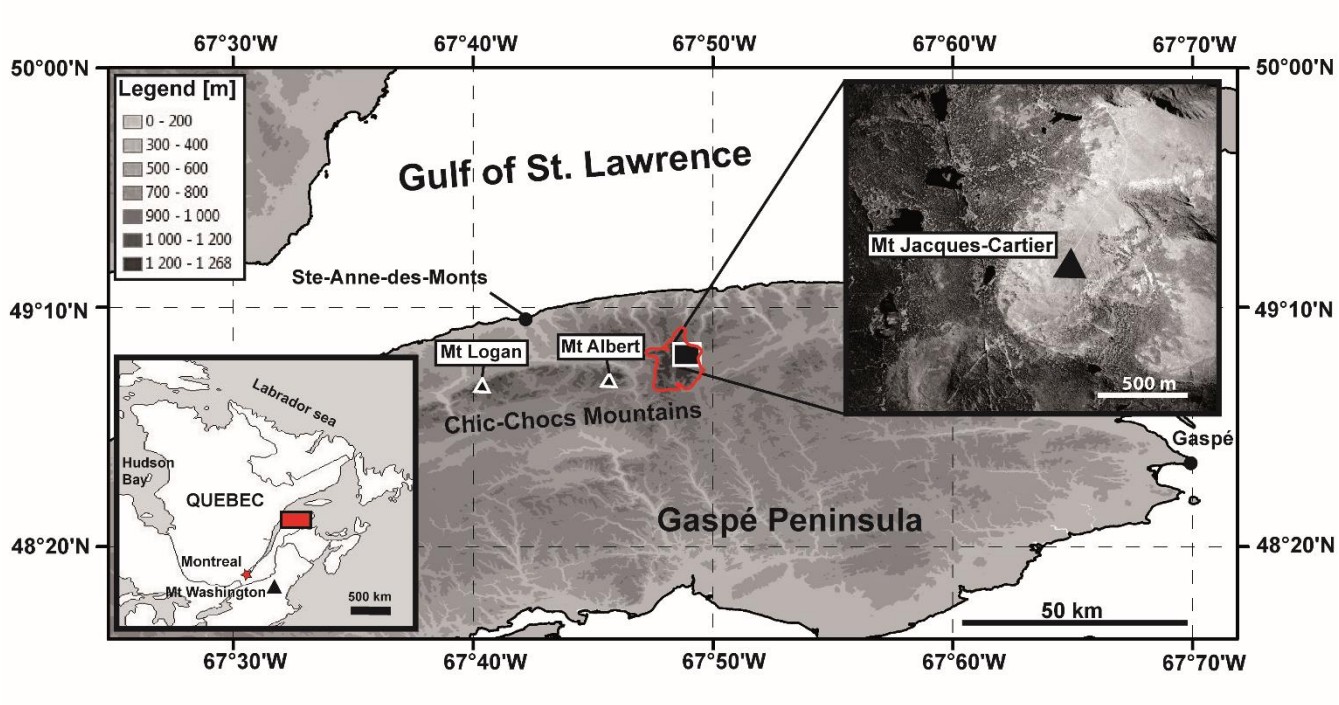

**Figure. 1. Study site. The Mont Jacques-Cartier (1268 m a.s.l.) is the highest summit of the Chic-Chocs Mountains. It is surrounded by several summits exceeding 1100 m *a.s.l*. e.g. Mont Logan (1150 m *a.s.l.*) and Mont Albert (1154 m *a.s.l.*). The red dashed line delineates the batholith of the McGerrigle Mountains. The Mont Jacques-Cartier summit forms a treeless dome above 1200 to 1220 m *a.s.l* where a typical alpine tundra environment is present.**

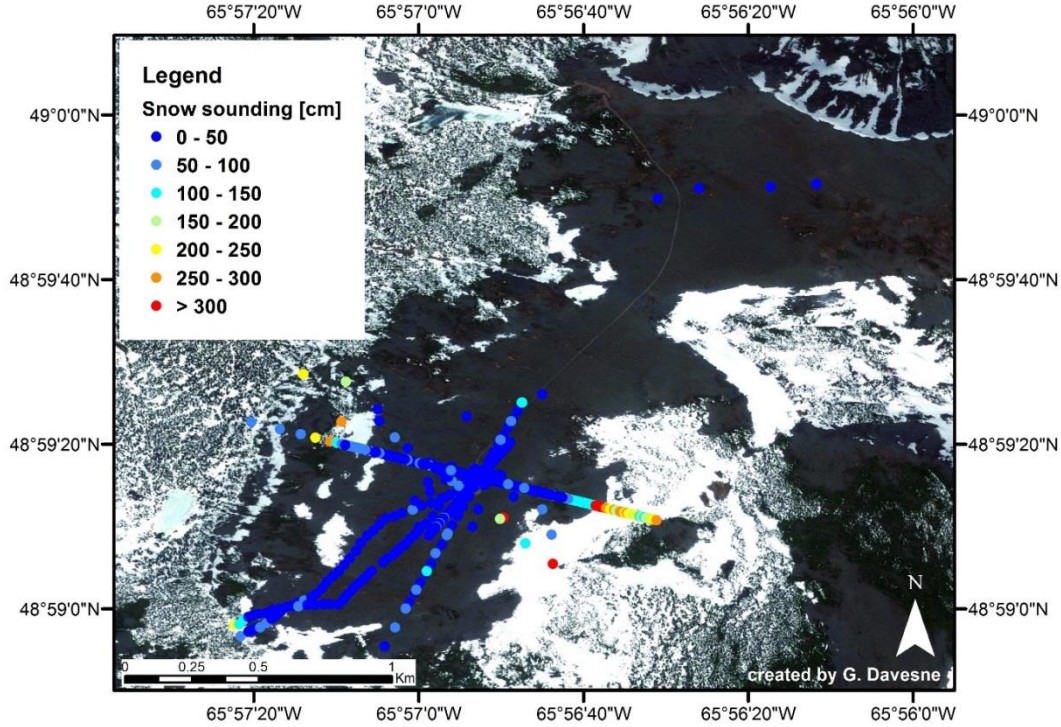

Figure 2. Compilation of snow measurements made in 1979, 1980, 2009, 2010, 2011, 2012, 2013 over the Mont Jacques-Cartier summit. The Spot-5 image in the background was taken on May 28th, 2013 (Google™ Earth, 2013) and thus shows only residual snow patches.

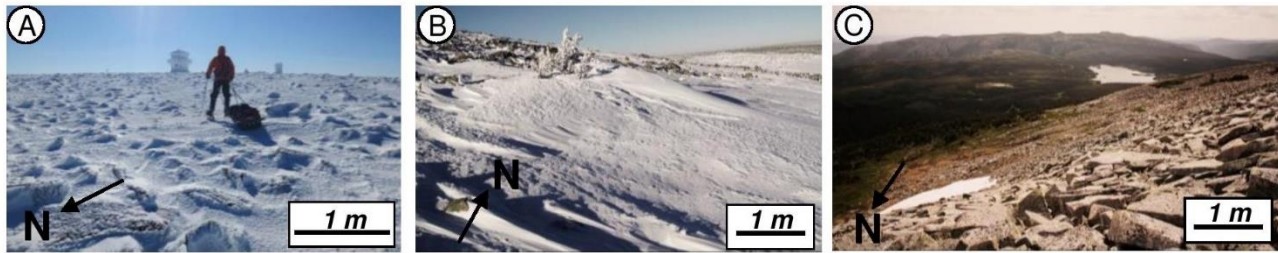

Figure 3. Photographs of Mont Jacques-Cartier. A) Blockfield after a snowstorm in early April 2014. Even at the end of the winter the large blocks were still protruding from the shallow snowpack; B) An isolated patch of krummholz on the southeast slope with typically leeward trailing snow accumulation zone (Feb. 2012); C) Long-lasting snow patch in the topographic depression of the leeward southeast slope at the end of July 2014.

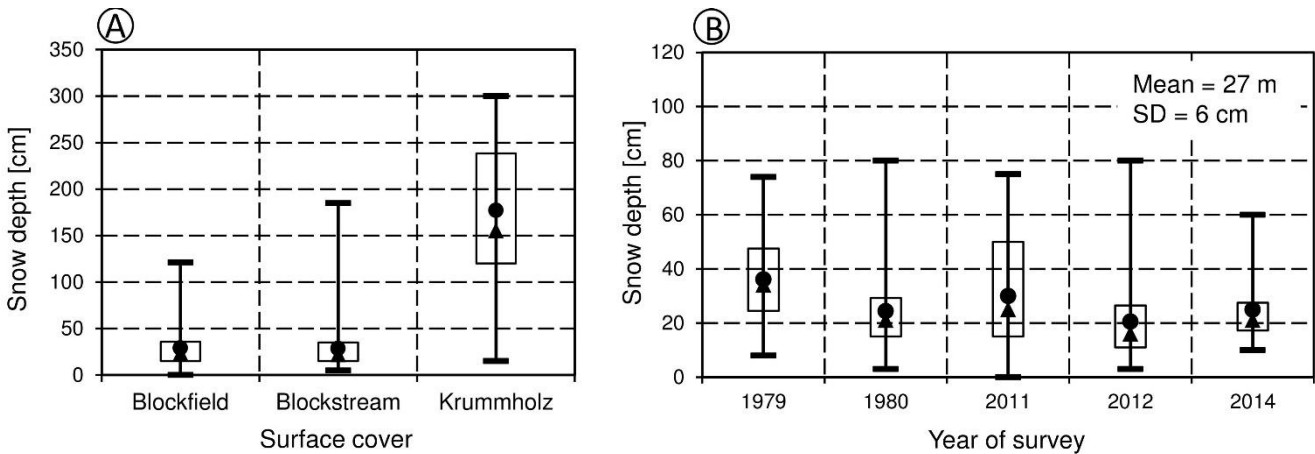

**Figure 4. Box-and-whisker plots showing (A) the uneven snow thickness between the alpine tundra environment (Blockfields and blockstreams) and the krummholz belt; and (B) the inter-annual variability of the snowpack thickness on the wind-swept summit. The "box" is delimited by the upper and the lower quartile; the median and the mean are represented by the triangle and the circle, respectively. The "whiskers" represent the maximum and minimum snow height values.**

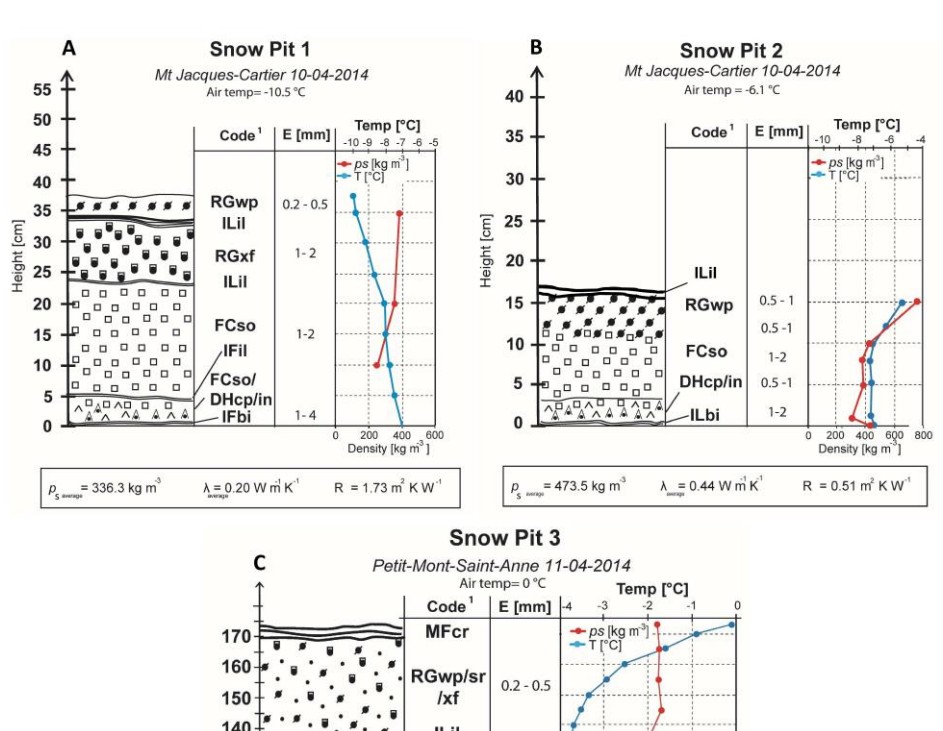

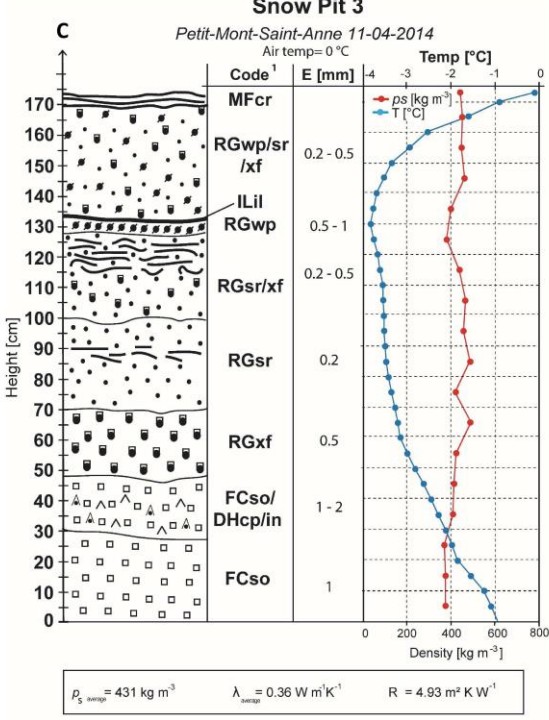

**Figure 5. Snow stratigraphy interpreted from profiles excavated in snowpacks on the tundra areas of Mont Jacques-Cartier (A and B) and on a krummholz patch on the Petit Mont Saint Anne summit (C) in April 2014. Code: RGwp: wind packed rounded grains; RGsr: small rounded grains; RGxf: faceted rounded particles; FCso: solid faceted crystals; MFcr: Melt-freeze crust; DHcp: depth hoar, hollow cups; ILil: horizontal ice layer (Fiez et al., 2009). DHin, indurated depth hoar, as defined in Domine et al. (2016b) and**
**whose symbol which combines a depth hoar symbol with a small rounded grain symbol, has been used here. $\rho_s$ (kg m$^{-3}$), $\lambda$ (W m$^{-1}$ K$^{-1}$) and R (m$^2$ K W$^{-1}$) represent respectively the density, the thermal conductivity and the thermal resistance of the snowpack. The $\rho_s$ and $\lambda$ for the snowpits are the average of values measured for each layer of the snowpacks.**

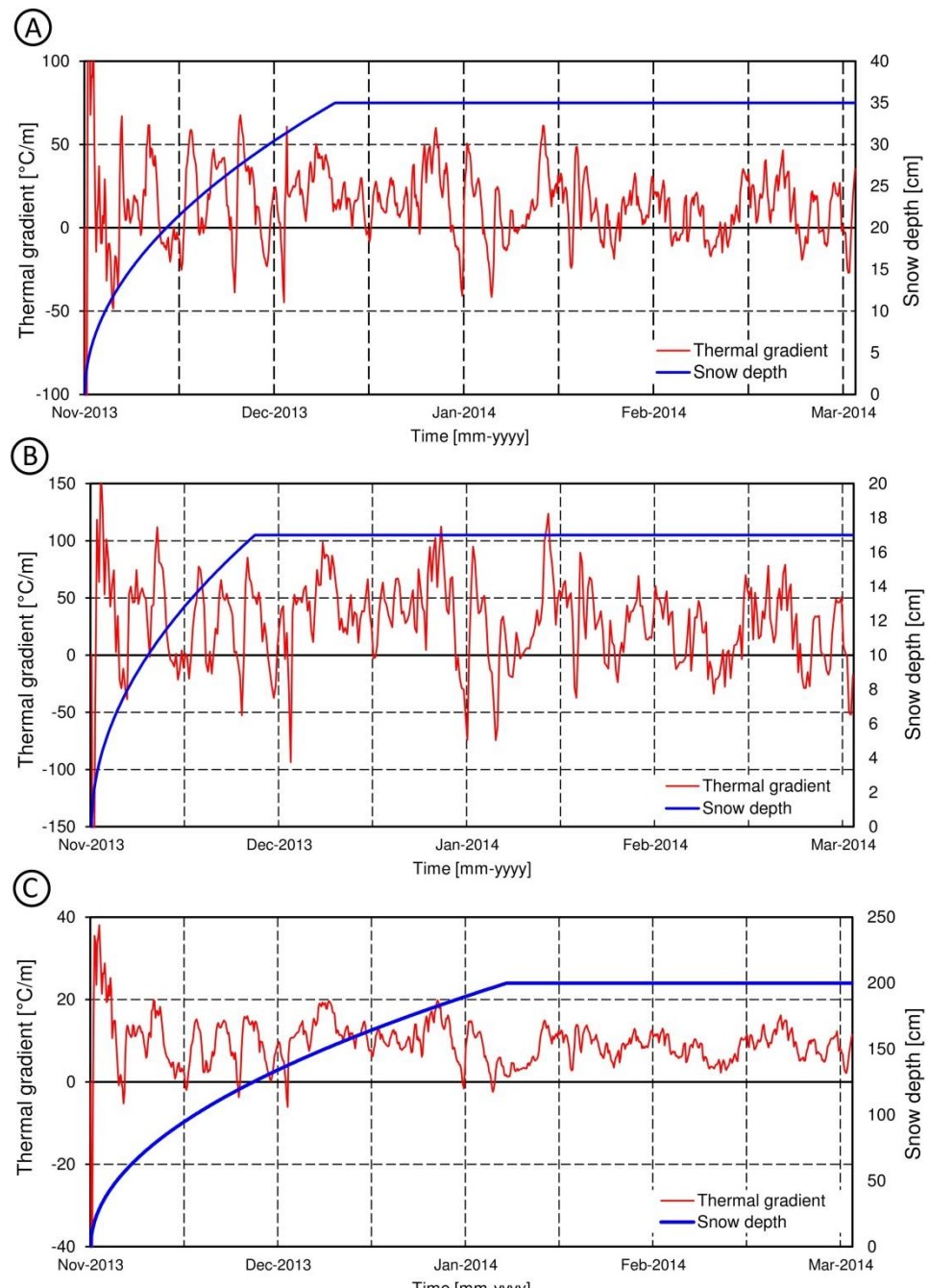

**Figure 6. Temperature gradient in the snowpack calculated through the winter for the 3 sites of snow pits presented in Figure 4 (A = snow pit 1; B = snow pit 2 and C = snow pit 3). The postulated snow depths evolution used to calculate the gradient are also shown.**

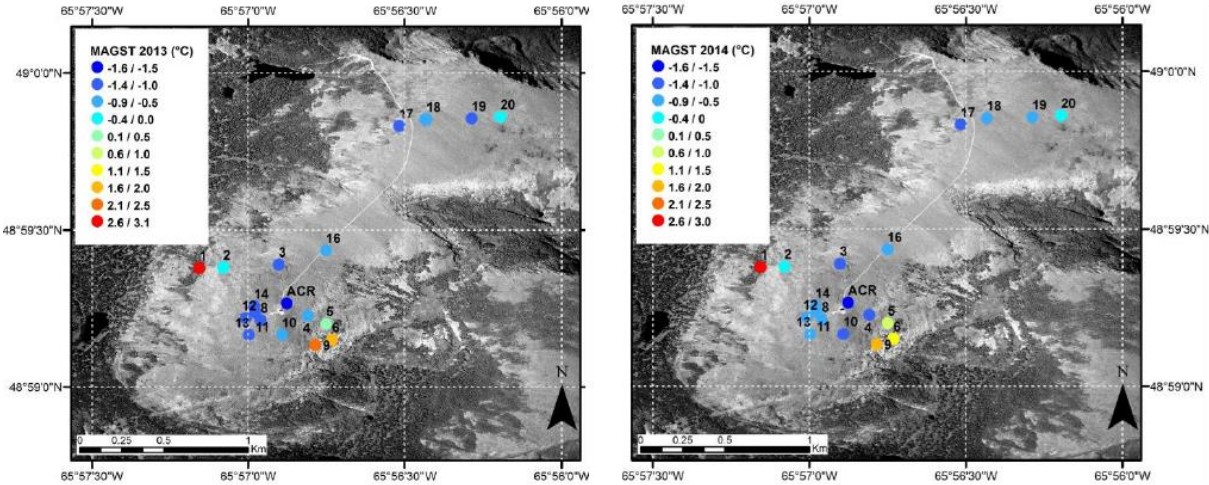

**Figure 7. Maps of the mean annual ground surface temperature (MAGST) recorded at the summit in 2013 and 2014.**

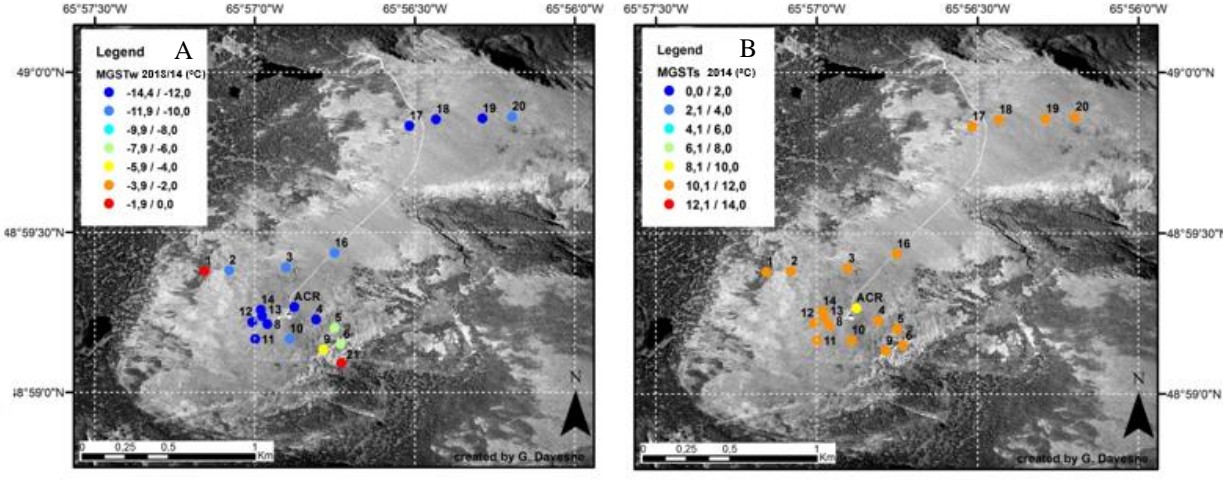

**Figure 8. Maps (A) of the mean winter ground surface temperature (MGST$_w$) measured by the sensors over the Mont Jacques-Cartier during the winter [Dec-Mar] 2013-2014, (B) of the mean summer ground surface temperature (MGST$_s$) measured in summer [Jun-Aug] 2014.**

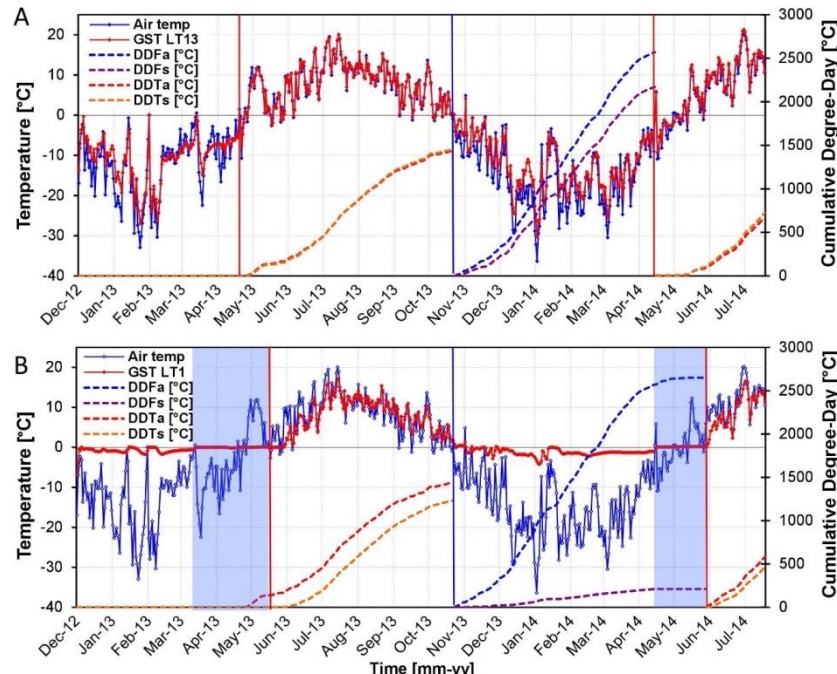

Figure 9. Evolution of the ground surface temperature (GST) from December 2012 to August 2014 for the sensors LT13 (A) and LT1 (B) which are representative of the thermal regime of the sensors on the zone with a thin snowpack and areas with a thick snowpack respectively. The dashed lines represent the cumulative freezing degree-day (DDF) and thawing degree-day (DDT) at the ground surface and in the air. The red vertical lines mark the end of the freezing season while the blue lines mark the beginning. Finally, the blue zones represent the duration of the zero curtain effect period.

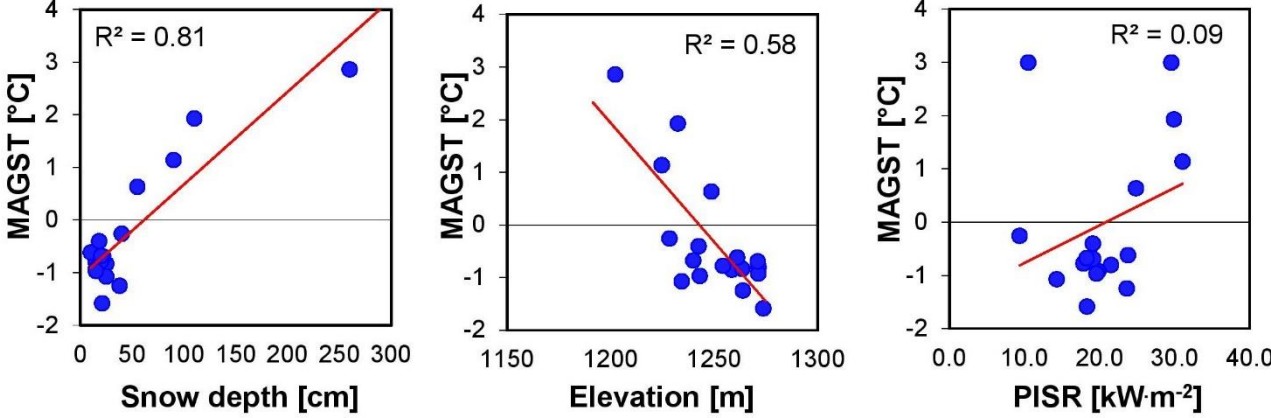

Figure 10. Linear regression line between the mean annual ground surface temperature (MAGST) in 2014 and snow depth, elevation and potential incoming solar radiation (PISR).

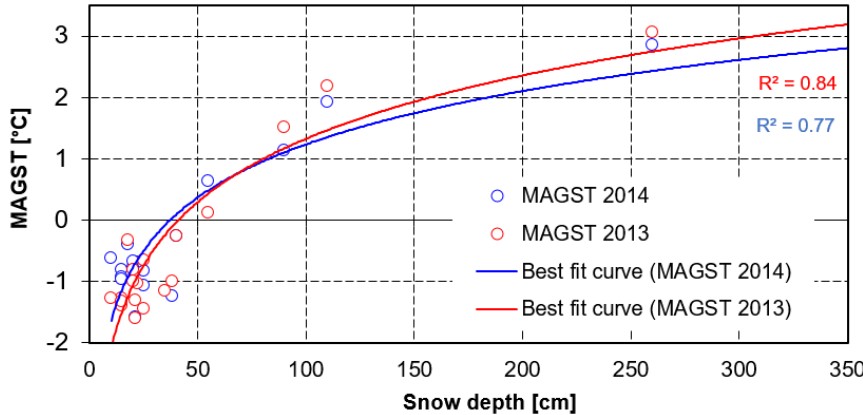

Figure 11. Relationship between the snow height measured in April 2014 and the mean annual ground surface temperature (MAGST) in 2013 and 2014.

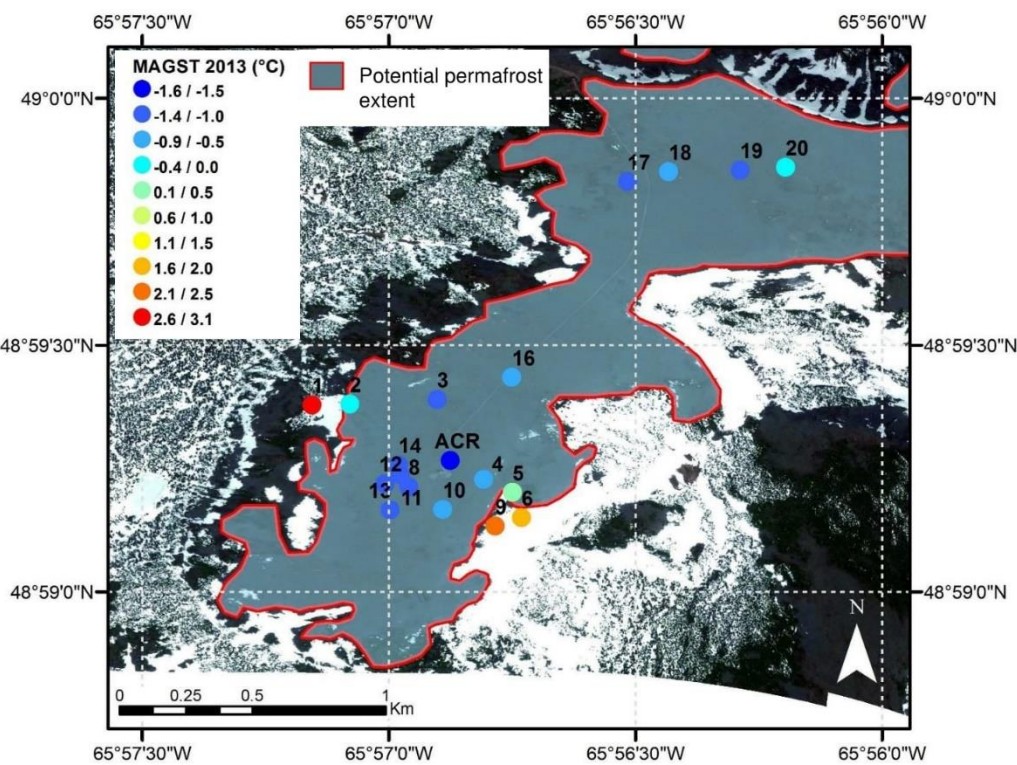

**Figure 12. Map of potential permafrost distribution based on the snow and vegetation distribution extracted from the satellite image Spot-5 taken on May 28[th] 2013. The mean annual ground surface temperatures (MAGST) of 2013 have been added for validation. Each sensor appears as a point and is labelled.**

| Air temp | $MAAT_{2013}$ | $MAAT_{2014}$ | $MAT_{w\ 2013/14}$ | $MAT_{w\ 2014/15}$ |
|---|---|---|---|---|
| | -2.27 | -2.77 | -18.05 | -17.65 |

| Zone | Sensors | $MAGST_{2013}$ | $MAGST_{2014}$ | $MGST_{w\ 2013/14}$ | $MGST_{w\ 2014/15}$ | $\Delta T_{w\ 2013/14}$ | $\Delta T_{w\ 2014/15}$ | $nf_{2013/14}$ | $nf_{2014/15}$ | $d_s$ [cm] |
|---|---|---|---|---|---|---|---|---|---|---|
| Wind exposed plateau | #2 | -0.2 | -0.3 | -11.3 | -9.7 | 6.8 | 7.9 | 0.66 | 0.61 | 40 |
| | #3 | -1.3 | -1.1 | -10.7 | -10.5 | 7.3 | 7.2 | 0.66 | 0.65 | 21 |
| | #4 | -1.0 | -1.2 | -13.4 | -12.3 | 4.6 | 5.3 | 0.76 | 0.74 | 38 |
| | #8 | -1.3 | -0.8 | -13.4 | -12.1 | 4.7 | 5.5 | 0.76 | 0.71 | 15 |
| | #10 | -0.6 | -0.9 | -11.5 | -11.3 | 6.5 | 6.4 | 0.68 | 0.72 | 25 |
| | #11 | -1.3 | -0.6 | -13.5 | -10.3 | 4.6 | 7.4 | 0.78 | 0.62 | 10 |
| | #12 | -1.1 | -0.8 | -13.3 | -12.0 | 4.8 | 5.7 | 0.77 | 0.71 | 35 |
| | #13 | -1.4 | -0.9 | -15.4 | -13.4 | 2.7 | 4.2 | 0.84 | 0.75 | 15 |
| | #14 | -1.0 | -0.7 | -13.6 | -13.2 | 4.5 | 4.4 | 0.78 | 0.76 | 22 |
| | #16 | -1.0 | -0.8 | -11.0 | -11.3 | 7.1 | 6.4 | 0.66 | 0.67 | 20 |
| | #17 | -1.4 | -1.1 | -15.4 | -10.7 | 2.6 | 6.9 | 0.86 | 0.67 | 25 |
| | #18 | -0.8 | -0.7 | -13.5 | -14.0 | 4.6 | 3.7 | 0.74 | 0.76 | 20 |
| | #19 | -1.3 | -1.0 | -14.6 | -13.4 | 3.4 | 4.3 | 0.79 | 0.77 | 15 |
| | #20 | -0.3 | -0.4 | -12.3 | -10.4 | 5.8 | 7.3 | 0.72 | 0.64 | 18 |
| | ACR | -1.6 | -1.6 | -13.0 | -12.8 | 5.1 | 4.9 | 0.74 | 0.74 | 21 |
| | Mean | -1.1 | -0.9 | -13.0 | -11.8 | 5.0 | 5.8 | 0.75 | 0.70 | 23 |
| | Std | 0.4 | 0.3 | 1.5 | 1.3 | 1.5 | 1.3 | 0.06 | 0.05 | 9 |
| Leeward slope | #5 | 0.1 | 0.6 | -7.5 | -10.0 | 10.6 | 7.7 | 0.48 | 0.62 | 60 |
| | #6 | 1.5 | 1.1 | -8.5 | -3.2 | 9.6 | 14.5 | 0.52 | 0.27 | 90 |
| | #9 | 2.2 | 1.9 | -4.9 | -3.7 | 13.2 | 14.0 | 0.31 | 0.29 | 110 |
| | #21 | - | - | - | -0.3 | - | 17.3 | - | 0.11 | >300 |
| | Mean | 1.3 | 1.2 | -6.9 | -4.3 | 11.1 | 13.4 | 0.4 | 0.3 | 87 |
| | Std | 1.1 | 0.7 | 1.9 | 4.1 | 1.9 | 4.1 | 0.1 | 0.2 | 25 |
| Krummholz | #1 | 3.1 | 2.9 | -1.2 | -0.9 | 16.9 | 16.8 | 0.08 | 0.10 | 260 |
| | Mean | - | - | - | - | - | - | - | - | - |
| | Std | - | - | - | - | - | - | - | - | - |

Table A1. Summary of the variable ground surface thermal conditions for each zone. The mean annual air temperature (MAAT) and mean annual ground surface temperature (MAGST) for 2013 and 2014 and the mean winter air temperature ($MAT_w$) and mean winter ground surface temperature ($MGST_w$) for both winters 2013-2014 and 2014-2015 have been calculated. From $MGST_w$ and the $MAT_w$, the average surface thermal offset ($\Delta T$) of both winters have been calculated. The freezing N-factor (nf) was calculated for the freezing season 2013-2014 and 2014-2015 and thawing N-factor (nt) was calculated for thawing season 2013 and 2014. The snow depth ($d_s$) was measured in April 2014.