# Peer review of "Wind-driven snow conditions control the occurrence of contemporary marginal mountain permafrost in the Chic-Chocs Mountains, southeastern Canada: a case study from Mont Jacques-Cartier"

_The Cryosphere, 2016_

## Referee Comment (RC1) · Anonymous Referee #1 · 31 Oct 2016

The manuscript by Davesne et al. investigates the impact of snow cover on permafrost distribution in the mountains of south-eastern Canada, focussing on Mont Jacques-Cartier. An impressive data set is utilized consisting of data collected over 4 decades along with more recently collected information on ground surface temperatures (GST). This study complements those done in western Canada in the southern Yukon (e.g. Lewkowicz et al. 2012; Bonnaventure et al. 2012) and in the European Alps (e.g. Gruber and Hoezle, 2001). The paper is fairly well written with the objectives clearly

articulated. Results and interpretations are effectively presented and the conclusions drawn from these are sound. There are no major concerns with the manuscript and with some minor revisions it should be acceptable for publication.

The paper might benefit from a comparison to some of the work done regarding snow cover and permafrost conditions across latitudinal treeline (e.g. Palmer et al. 2012). Also, much work has been done with respect to permafrost mapping utilizing the BTS (basal temperature of snow) approach in the Swiss Alps and also more recently in northwestern Canada. It might be useful to compare the equilibrium winter GST obtained for Mont Jacques-Cartier to the range in BTS utilized to determine permafrost probability in these other studies. Do you get the same threshold values etc.

Additional comments and suggestions are provided below for consideration in preparation of the revised manuscript. Many of these are suggestions for editorial revisions. The authors also need to check some of their figure and table references. Only a minimal amount of work should be required to address the comments and prepare a revised manuscript.

Specific comments (keyed to line number) L13 – You could mention the type of data utilized in your analysis to demonstrate your hypothesis (rather than outlining objectives in the next sentence).

L15 replace "was" with "were" at end of line

L25-29 – Additional papers that may be relevant here and elsewhere in the paper that have considered permafrost in mountains in western Canada: Lewkowicz et al. (2012); Bonnaventure et al. (2012).

L30-31 – It is not clear here what you mean by snow cover providing a cooling or warming effect. Do you mean if there is little snow, then greater heat loss occurs so surface temperatures will be lower. Also are you referring to the "surface offset" – see Smith and Riseborough (2002)

L41 – revision suggested "..and the spatial extent of . . .." OR say "spatial distribution of permafrost at this site"

L48 – Do you mean surface offset? See Smith and Riseborough (2002)

L71 – Should it be "surface environmental lapse rate"

L91 – Revision suggested ". . .deep temperature cable that has been monitored continuously since 1977.."

L93 – You could say "Early measurements between 1977 and x, indicated . . .." (I assume since you give a range that these are measurements made over a few years)

L94-99 – give temperature at ZAA at beginning of the monitoring period so comparison can be made with the 2013 value. Also why not just say that the temperature at ZAA had risen to -0.3°C by 2013 indicating warming and degradation of permafrost.

L101 – Isn't the impact of snow on GST fairly well known from other studies?

L103 – "measured" might be better word than "monitored"

L110-111 – suggested revisions ". . .probe 350 mc long." ". . .generally conducted in (late?) March or early April.."

L121 – refer to Fig. 1 for location of Petit Mont Saint-Anne

L122 – "determine" might be better word than "measure" since some things are calculated from measured values

L131 – Isn't Lunardini (1981) the original reference for this?

L136 – Domine et al. (2011) is not in reference list – right year?

L142 – do you mean complete melt/disappearance of snow pack?

L144 – Table 2? (gives sensor location)

L147 – Positive air or surface temperature?

L153 – Table 2? Do you mean "beneath a deep snow-bank"?

L162 – suggested revision "...on the MAGST was assessed using..."

L162-163 – do you mean 2013-14 MAGST?

L170-175 – Did you define the freezing season using the GST and use the same period for summing the air and surface freezing degree days. This is what was done by Karunaratne and Burn (2003). Others (e.g. Lewkowicz et al. 2012) consider the air and surface freezing season separately (i.e. summing freezing degree days over different periods so that for the air this will be the total FDD not just during the period where GST<0°C). You could also cite the CJES paper for Karunaratne and Burn (2004).

L176-178 – Some of the literature related to BTS (Basal temperature of snow) might be relevant here.

L178-179 – Latent heat is also released as the active layer freezes in the fall and winter and this can maintain GST near 0°C – see for example Riseborough and Smith (1998)

Results section – In some places results seem to be combined with background information and some interpretation that might be better in the discussion section.

L200 – suggested revision "..depth was greater than..."

L211 – suggested revision "...was similar to that observed...."

L222 – Do you mean Fig 5b? Also, you need to label a,b,c on the figure.

L251 – Elevation linked to air temperature, vegetation influence?

L255 – suggested revision "...was highly spatially variable..."

L256 – You could say there is a range in winter GST of 14°C. Also you should refer to "sites" rather than "sensors".

L263-264 – Heat is conducted but not temperature. Why don't you just say that there is limited insulation provided by the snow pack.

L269 – Beneath the snow bank

L289-293 – You probably don't need this.

L297 – replace "was" with "were"

L322 – Can you comment on what this means for GST?

L328- Delete last part of sentence regarding giving names to figs.

L330-355 – You could write in a more passive voice in this section.

L341 – "were" is probably better than "are"; "in" is probably better than "on" L342 – "fluctuated" might be better word

L349 – revise "... high values up to 100"

L377 – This short zero curtain might also be related to limited moisture content of the active layer (rapid freeze back and minimal latent heat effect) - see for eg. Riseborough 2001; Riseborough and Smith 1998.

Figures Fig. 2 – Avg for all years? Fig. 3 – These figures are labelled as 1,2,3 so they could be referred to that way in the caption. Fig. 4 – Would it be possible to have a similar type of figure but instead of year of survey you could use veg type/surface cover instead. You could also have a similar figure for GST. Fig. 5 – Label A,B C as this is how you refer to them in the text. Fig. 7 – It would be better to say that this shows the best fit regression line rather than the correlation. Fig. 12 – Correct legend – replace "extend" with "extent"

Other references (not in reference list) for the authors' consideration Bonnaventure, P. P., Lewkowicz, A. G., Kremer, M., and Sawada, M. C., 2012: A permafrost probability model for the southern Yukon and northern British Columbia, Canada. Permafrost and Periglacial Processes, 23: 52-68.

Bonnaventure, P. P. and Lewkowicz, A. G., 2012: Permafrost probability modeling

above and below treeline, Yukon, Canada. Cold Regions Science and Technology, 79-80: 92-106 Karunaratne, K. C. and Burn, C. R., 2004: Relations between air and surface temperatures in discontinuous permafrost terrain near Mayo, Yukon Territory. Canadian Journal Earth Sciences, 41: 1437-1451.

Lewkowicz, A. G., Bonnaventure, P. P., Smith, S. L., and Kuntz, Z., 2012: Spatial and thermal characteristics of mountain permafrost, northwest Canada. Geografiska Annaler: Series A Physical Geography, 94: 195-215.

Palmer, M. J., Burn, C. R., and Kokelj, S. V., 2012: Factors influencing permafrost temperatures across tree line in the uplands east of the Mackenzie Delta, 2004–2010. Canadian Journal of Earth Sciences, 49: 877-894.

Riseborough, D. W., 2001: An analytical model of the ground surface temperature under snowcover with soil freezing. 58th Annual Eastern Snow Conference: 363-372.

Riseborough, D. W. and Smith, M. W., 1998: Exploring the limits of permafrost. Proceedings of Seventh International Conference on Permafrost, 57: 935-941.

Smith, M. W. and Riseborough, D. W., 2002: Climate and limits of permafrost: a zonal analysis. Permafrost and Periglacial Processes, 13: 1-15.

---

## Referee Comment (RC2) · Anonymous Referee #2 · 18 Nov 2016

In the manuscript (MS), Davesne et al. explored how the thermo-physical properties of snow affect the spatial distribution of ground surface temperatures and permafrost on the summit of Mont Jacques-Cartier, Canada. The authors used extensive field data on snow properties and ground temperatures. Based on the results, snow conditions controlled the small-scale spatial variability of ground surface temperatures. The snow properties were determined by wind conditions, local topography, surface roughness and vegetation cover. The results of this study are useful in the exploration of per-

mafrost in alpine environments, where permafrost can be marginal and thus sensitive to the changes in environmental conditions. Consequently, the topic of the MS fits well to The Cryosphere.

In general, I consider this MS to be a good contribution in investigating snow effects on ground temperatures. However, I have one major and several smaller suggestions to improve the presentation. An additional concern is the rather local and descriptive nature of the paper i.e. is the MS innovative enough for the journal? I suppose the novelty value is not a critical issue and the MS is suitable for the planned special sections of the journal. After all, I recommend publication of the MS in The Cryosphere after moderate/major revision.

Major comment: My major concerns are related to the results section 4.3 and discussion section 5.2 and partly 5.3. The first paragraph of the section 4.3 is a mixture results and discussion. Thus, it is somehow difficult to be sure which results are from this study and which are derived from the literature. On the contrary, the sections 5.2 and 5.3 (lines 385-390) included completely new results. To my opinion, the above mentioned sections of the discussions should be in the results.

Specific comments:

Title: Why is there a full stop in the end?

Abstract: The abstract is partly incomplete. It presents the aims and results but lack conclusions.

Line 13: It would be nice to see the absolute elevation of the studied mountain (in the brackets after the name).

Line 20 and 23: Please be consistent in the use of space between numbers and °C. Moreover, use minus sign instead of soft hyphen (-) in relevant places throughout the MS.

Line 31: To my opinion, the Table 1 is not needed and could be deleted because there

already are many tables and figures in the MS (and Table 1 is the first to remove).

Lines 37-38 (Howe, 1971): Can the presence of permafrost be based on an over 40 year old reference in this marginal permafrost environment (especially considering what is presented in lines 96-99)?

Line 49: Spell out MAGST.

The section 2: Relative elevations could be presented somewhere (relevant when considering temperature inversions).

Line 90: A bracket missing.

Lines 108-109: How typical were the meteorological conditions of the studied years compared to the long-term climate conditions (based on data from the nearest met station)?

Line 116: Why didn't you use freely available Landsat scenes of the study years to explore the general patterns of snow ablation and accumulation?

Line 144: Reference to a wrong table? Also line 153.

Lines 193, 196 and 199: I think "Fig. 3, Photo 1" could be "Fig. 3A" etc.?

Line 198: Gelifluction? Or rather solifluction (gelifluction + frost creep) in this environment?

Line 261: Rather alpine than tundra (please check and be consistent throughout the MS).

Line 274: Amazingly low minimum temperature considering the measurement site (summit and ground surface)?

Lines 419-422: It would be nice to see a bit more discussion on this topic (the results of this study–sensitivity of marginal permafrost–climate change indicator)

Conclusions: In the end of this section, there could be a more general conclusion(s) of

the study results–permafrost sensitivity–climate change interface.

References: I recommend the authors to carefully check the list for minor inconsistency (e.g. full journal names vs. short versions).

Table 4: Please spell out/explain abbreviations and symbols. Tables and figures should stand alone.

Figure 5: Please spell out/explain abbreviations and symbols (ps, lambda and R).

Figure 6: Please spell out MAGST and ACR and explain the meaning of numbered locations (also Fig. 8). Why there are two values separated by backslash in the legend (check also other comparable figures)?

Figure 7: Please spell out MAGST. How was the PISR variable computed (not mentioned in the methods section)?

Figure 9: Please spell out GST, DDF and DDT. Maybe LTs should be explained as well?

Figures 11 and 12: Please spell out MAGST.

Appendix A: Please spell out abbreviations.

––––––––––––––––––––––––––––––

---

## Author Comment (AC1) · 9 Jan 2017

We would like to thank Anonymous Referee #1 for the constructive review which permitted to improve the manuscript. His/her careful reading of the manuscript and his/her good knowledge of the subject-matter allowed providing relevant suggestions and additions to the manuscript. We treat each point raised in detail and with great interest. Note that the line numbers given in this response refer to the revised version of the manuscript in track changes mode.

[Figure]

General comments: Comment 1: Referee #1: Much work has been done with respect to permafrost mapping utilizing the BTS (basal temperature of snow) approach in the Swiss Alps and also more recently in northwestern Canada. It might be useful to compare the equilibrium winter GST obtained for Mont Jacques-Cartier to the range in BTS utilized to determine permafrost probability in these other studies.

Authors: We agree with the importance of mentioning studies undertaken in other sites (especially in the European Alps, the Canadian Rocky Mountains, and the Japanese Alps) regarding the BTS technique used to predict the presence and absence of permafrost. We therefore added complementary information in the introduction (lines 44 to 52) to briefly describe the BTS technique and its limitations for our study. Indeed, on the plateau of Mont Jacques-Cartier - where the snow thickness is very low - the BTS technique is not applicable because a snowpack of at least 80 cm is required to produce reliable BTS values to predict permafrost occurrence. We thus choose to install temperature data loggers to continuously measure the GST for 2 years which enabled determining the threshold of snow thickness above which the MAGST exceeds 0°C. As suggested by the Referee #1, we compare the threshold found for Mont Jacques-Cartier with the one found by Smith and Riseborough (2002) in the discussion part (lines 443 to 447).

Specific comments:

Comment 1: Referee #1: L13 – You could mention the type of data utilized in your analysis to demonstrate your hypothesis (rather than outlining objectives in the next sentence).

Authors: we agree and modified the beginning of the abstract (Lines 14-15).

Comment 2: Referee #1: L15 replace "was" with "were" at end of line

Authors: modification made

Comment 3: Referee #1: L25-29 – Additional papers that may be relevant here and

elsewhere in the paper that have considered permafrost in mountains in western Canada: Lewkowicz et al. (2012); Bonnaventure et al. (2012).

Authors: We agree. We added Harris (1981) and Lewkowicz et al. (2012) and Bonnaventure et al. (2012) as suggested by the reviewer. We also added Magnin et al., 2016 as the most recent reference available for the Alps regarding the study of the snow control on permafrost (Lines 35-37).

Comment 4: Referee #1: L30-31 – It is not clear here what you mean by snow cover providing a cooling or warming effect. Do you mean if there is little snow, then greater heat loss occurs so surface temperatures will be lower. Also, are you referring to the "surface offset" – see Smith and Riseborough (2002).

Authors: Yes, the thermal effect brought by a snowpack depends on the balance between cooling and warming effects, whose magnitude depends in turn on the thickness, duration, timing and thermal and optical properties of the snow. The warming effect is mainly brought by the insulating capacity of the snow while the cooling effect is brought by the albedo of snow for short wave radiation and its melting which favours latent heat consumption and thus delayed ground surface warming. We agree to use the term "surface thermal offset" to be consistent with Smith and Riseborough (2002) to qualify the offset brought by the snowpack between air temperature and GST. Modifications have been made through the entire MS.

Comment 5: Referee #1: L41 – revision suggested "..and the spatial extent of :" OR say "spatial distribution of permafrost at this site".

Authors: We modified for "..and the spatial extent of".

Comment 6: Referee #1: L48 – Do you mean surface offset? See Smith and Riseborough (2002)

Authors: Yes, modification made to be consistent with Smith and Riseborough (2002).

Comment 7: Referee #1: L71 – Should it be "surface environmental lapse rate"

Authors: We agree. The term environmental lapse rate is indeed more adapted in this case.

Comment 8: Referee #1: L91 – Revision suggested ": : :deep temperature cable that has been monitored continuously since 1977.."

Authors: revision accepted

Comment 9: Referee #1: L93 – You could say "Early measurements between 1977 and x, indicated : : :." (I assume since you give a range that these are measurements made over a few years)

Authors: No, we mentioned here the first measurements made in 1977 following the thermistor cable installation. We made slight changes to clarify this part (lines 119-120).

Comment 10: Referee #1: L94-99 – give temperature at ZAA at beginning of the monitoring period so comparison can be made with the 2013 value. Also why not just say that the temperature at ZAA had risen to -0.3_C by 2013 indicating warming and degradation of permafrost.

Authors: We agree, modification made line 123.

Comment 11: Referee #1: L101 – Isn't the impact of snow on GST fairly well known from other studies?

Authors: Yes, many studies already dealt with the impact of snow on the ground surface thermal regime. We changed the sentence to be more specific on the study case of Mont Jacques-Cartier.

Comment 12: Referee #1: L103 – "measured" might be better word than "monitored"
Authors: We agree

Comment 13: Referee #1: L110-111 – suggested revisions ": : :probe 350 mc long." ": : :generally conducted in (late?) March or early April.."

Authors: suggestion accepted Comment 14: Referee #1: L121 – refer to Fig. 1 for location of Petit Mont Saint-Anne

Authors: Fig. 1 added

Comment 15: Referee #1: L122 – "determine" might be better word than "measure" since some things are calculated from measured values.

Authors: suggestion accepted

Comment 16: Referee #1: L131 – Isn't Lunardini (1981) the original reference for this? Authors: Yes, the original reference is indeed Lunardini (1981). Modification made.

Comment 17: Referee #1: L136 – Domine et al. (2011) is not in reference list – right year?

Authors: Reference added in the list

Comment 18: Referee #1: L142 – do you mean complete melt/disappearance of snow pack?

Authors: Yes, clarification made

Comment 19: Referee #1: L147 – Positive air or surface temperature?

Authors: Positive air temperature. Modification made.

Comment 20: Referee #1: L153 – Do you mean "beneath a deep snow-bank"?

Authors: Yes, modification made.

Comment 21: Referee #1: L162 – suggested revision ": : :on the MAGST was assessed using: : :"

Authors: we agree, suggestion accepted.

Comment 22: Referee #1: L170-175 – Did you define the freezing season using the GST and use the same period for summing the air and surface freezing degree days.

This is what was done by Karunaratne and Burn (2003). Others (e.g. Lewkowicz et al. 2012) consider the air and surface freezing season separately.

Authors: Yes, we considered the air and surface temperature separately. We made slight changes in the manuscript to better distinguish the air and surface freezing DD.

Comment 23: Referee #1: L176-178 – Some of the literature related to BTS (Basal temperature of snow) might be relevant here.

Authors: We added Hoelzle et al (1992); Ishikawa, (2003) as reference related to the concept of WEqT and BTS (line 230).

Comment 24: Referee #1: L178-179 – Latent heat is also released as the active layer freezes in the fall and winter and this can maintain GST near 0_C – see for example Riseborough and Smith (1998).

Authors: Yes, that is true, but in Mont Jacques-Cartier the water content of the regolith layer which covered the summit is expected to be very low. The zero-curtain effect brought by the freezing of the active layer is thus very limited and is not detectable on the GST recorded over the plateau.

Comment 25: Referee #1: Results section – In some places results seem to be combined with background information and some interpretation that might be better in the discussion section.

Authors: We agree with this comment. The section 4.3 Snow physical and thermal properties was a mixture of results and interpretation, while the section 5.2 Metamorphism and physical properties of the snowpack incorporated new results. As suggested by the referee 2, we reworked deeply both sections. In the section 4.3. (Results), the first paragraph was moved to the section 5.2. (Discussion) lines 380 to 393. In the section 5.2 (Discussion), we moved the paragraph which explains how we calculated the thermal gradient through the snow pack – to the methodology (lines 170 to 183). We also moved the results of the thermal gradient calculation to the section 4.3. (Results)

[Figure]

(lines 281 to 285 and lines 293 to 295). Figure 10 became figure 6.

Comment 26: Referee #1: L200 – suggested revision "..depth was greater than: : :"

Authors: Suggestion accepted

Comment 27: Referee #1: L211 – suggested revision ": : :was similar to that observed: : :"

Authors: Suggestion accepted

Comment 28: Referee #1: L222 – Do you mean Fig 5b? Also, you need to label a,b,c on the figure.

Authors: Yes, modification made.

Comment 29: Referee #1: L251 – Elevation linked to air temperature, vegetation influence?

Authors: Unfortunately, we cannot statically study the influence of the vegetation on the GST because only 1 sensor is installed on the krummholz belt.

Comment 30: Referee #1: L255 – suggested revision ": : :was highly spatially variable: : :" Authors: Suggestion accepted

Comment 31: Referee #1: L256 – You could say there is a range in winter GST of 14_C. Also you should refer to"sites" rather than "sensors".

Authors: Yes, modification made.

Comment 32: Referee #1: L263-264 – Heat is conducted but not temperature. Why don't you just say that there is limited insulation provided by the snow pack.

Authors: We agree. We removed the unclear sentences.

Comment 33: Referee #1: L269 – Beneath the snow bank

Authors: Suggestion accepted
Comment 34: Referee #1: L289-293 – You probably don't need this.

Authors: We preferred to keep that sentence because it enables to clearly introduce the various parts of the discussion.

Comment 35: Referee #1: L328- Delete last part of sentence regarding giving names to figs.

Authors: We agree.

Comment 36: Referee #1: L330-355 – You could write in a more passive voice in this section.

Authors: We agree. This section has been reworked.

Comment 37: Referee #1: L341 – "were" is probably better than "are"; "in" is probably better than "on" L342 –"fluctuated" might be better word.

Authors: Yes, modification made.

Comment 38: Referee #1: L349 – revise ": : : high values up to 100"

Authors: modification made.

Comment 39: Referee #1: L377 – This short zero curtain might also be related to limited moisture content of the active layer (rapid freeze back and minimal latent heat effect) - see for eg. Riseborough 2001; Riseborough and Smith 1998.

Authors: Yes, indeed, please see response to comment 23.

Additional changes:

The title 3.1. have been changed to Interannual snow thickness The title 3.3. have been changed to Snowpack onset and melt analysis The title 4.2. have been changed to Snowpack onset and melt

Please also note the supplement to this comment:

http://www.the-cryosphere-discuss.net/tc-2016-211/tc-2016-211-AC1-supplement.zip

---

## Author Comment (AC2) · 9 Jan 2017

We would like to thank Anonymous Referee #2 for the constructive review which permitted to improve the manuscript. His/her careful reading of the manuscript and his/her good knowledge of the subject-matter allowed to provide relevant suggestions and additions to the manuscript. We treat each point raised in detail and with great interest. Note that the line numbers given in this response refer to the revised version of the manuscript in track changes mode.

[Figure]

General comments: Comment 1: Referee #2: Is the MS innovative enough for the journal?

Authors: We are aware that the impact of the snow on permafrost thermal regime and distribution has already been studied in several sites around the planet, but not in eastern North America. At a regional scale, this study is therefore of great interest by providing a quantitative and qualitative understanding of the snow cover properties and effects on the ground surface thermal regime and mountain permafrost distribution in the Chic-Chocs Mountains and, most widely, in the Appalachian Range.

Comment 2:

Referee #2: Major comment: My major concerns are related to the results section 4.3 and discussion section 5.2 and partly 5.3. The first paragraph of the section 4.3 is a mixture results and discussion. Thus, it is somehow difficult to be sure which results are from this study and which are derived from the literature. On the contrary, the sections 5.2 and 5.3 (lines 385-390) included completely new results.

Authors: We agree with this comment. The section 4.3 Snow physical and thermal properties was a mixture of results and interpretation, while the section 5.2 Metamorphism and physical properties of the snowpack incorporated new results. As suggested by the referee 2, we reworked deeply both sections. In the section 4.3. (Results), the first paragraph was moved to the section 5.2. (Discussion) lines 380 to 393. In the section 5.2 (Discussion), we moved the paragraph which explains how we calculated the thermal gradient through the snow pack – to the methodology (lines 170 to 183). We also moved the results of the thermal gradient calculation to the section 4.3. (Results) (lines 281 to 285 and lines 293 to 295). Figure 10 became figure 6.

Specific comments:

Comment 3: Referee #2: Title: Why is there a full stop in the end?

Authors: We removed the full stop.

[Figure]

Comment 4: Referee #2: Abstract: The abstract is partly incomplete. It presents the aims and results but lack conclusions.

Authors: Agreed, we added a sentence that highlights the conclusion of the study (line 27 to 30).

Comment 5: Referee #2: Line 13: It would be nice to see the absolute elevation of the studied mountain (in the brackets after the name).

Authors: Agreed, we added the elevation (line 15).

Comment 6: Referee #2: Line 20 and 23: Please be consistent in the use of space between numbers and °C. Moreover, use minus sign instead of soft hyphen (-) in relevant places throughout the MS.

Authors: Agreed, modifications made to be consistent in the MS.

Comment 7: Referee #2: Line 31: To my opinion, the Table 1 is not needed and could be deleted because there already are many tables and figures in the MS (and Table 1 is the first to remove).

Authors: We consider this table to be useful for readers who are not familiar with the abbreviation regarding thermal terms.

Comment 8: Referee #2: Lines 37-38 (Howe, 1971): Can the presence of permafrost be based on an over 40 year old reference in this marginal permafrost environment (especially considering what is presented in lines 96-99)?

Authors: We cited the paper of Walegur and Nelson (2003). This reference, more recent than Howe (1971), confirms the present-day occurrence of permafrost in Mount Washington (line 55).

Comment 9: Referee #2: The section 2: Relative elevations could be presented somewhere (relevant when considering temperature inversions).

Authors: Agreed, we added the elevation for Cap-Chat and Cap-Madeleine weather stations (line 95).

Comment 10: Lines 108-109: How typical were the meteorological conditions of the studied years compared to the long-term climate conditions (based on data from the nearest met station)?

Authors: Unfortunately, the measurement of snow falls at the stations of Cap-Madeleine and Cap-Chat are discontinuous, consequently, we cannot calculate the annual total snow accumulations.

Comment 11: Referee #2: Line 116: Why didn't you use freely available Landsat scenes of the study years to explore the general patterns of snow ablation and accumulation?

Authors: A student in our lab made the study of the onset and melt dates of the seasonal snowpack over Mont Jacques-Cartier by analysing Landsat 5 and 7 images from 1990 to present. This study shows interesting results but the error was high due to the poor resolution of images, the frequent clouds cover which reduce the visibility of the target and the long lapse of time between 2 successive images. For this reason, we only deduced the timing and duration of the snowpack based on the daily GST recorded from 2008 to present at the borehole of Mont Jacques-Cartier.

Comment 12: Referee #2: Line 144: Reference to a wrong table? Also line 153.

Authors: Yes, corrections made.

Comment 13: Referee #2: Lines 193, 196 and 199: I think "Fig. 3, Photo 1" could be "Fig. 3A" etc.?

Authors: We agree, the figure 3, caption and citation in the MS have been modified as suggested.

Comment 14: Referee #2: Line 198: Gelifluction? Or rather solifluction (gelifluction +

frost creep) in this environment?

Authors: Yes, we agree that gelifluction is not the unique process which leads to the development of the solifluction lobe on the SE slope of Mont JC. The melt water derived from the long-lasting snowbank is likely the most important factor. We thus replaced gelifluction by solifluction in the MS (line 251).

Comment 15: Referee #2: Line 261: Rather alpine than tundra (please check and be consistent throughout the MS).

Authors: We agreed. We replace "tundra zone" by "alpine tundra zone" in the MS.

Comment 16: Referee #2: Line 274: Amazingly low minimum temperature considering the measurement site (summit and ground surface)?

Authors: The air temperature can drop below – 35 °C in winter at this elevation. A value of – 30 °C at the ground surface is thus not surprising in areas where the buffer effect played by the snowpack is very weak.

Comment 17: Referee #2: Lines 419-422: It would be nice to see a bit more discussion on this topic (the results of this study–sensitivity of marginal permafrost–climate change indicator).

Authors: We agree with this comment. We added the lines 484 to 489 to mention the high sensitivity of this kind of permafrost to the climate changes due to the quasi direct connexion between the air temperature and the internal ground temperature (no buffer effect played by snow, high thermal conductivity and low ice content of the bedrock). More information concerning the permafrost evolution and sensitivity in the recent context of climate change is available in the following paper we recently published, and to which we refer the reader

Gray, J.T., Davesne, G., Godin, E. and Fortier, D.: The Thermal Regime of Mountain Permafrost at the Summit of Mont Jacques‐Cartier in the Gaspé Peninsula, Québec, Canada: A 37 Year Record of Fluctuations showing an Overall Warming

[Figure]

Trend, Permafrost and Periglacial Processes. doi: 10.1002/ppp.1903, 2016.

Comment 18: Referee #2: Conclusions: In the end of this section, there could be a more general conclusion(s) of the study results–permafrost sensitivity–climate change interface.

Authors: We added the lines 409 to 515 to bring more general conclusions.

Comment 19: Referee #2: Table 4: Please spell out/explain abbreviations and symbols. Tables and figures should stand alone.

Authors: Modifications made.

Comment 20: Referee #2: Figure 5: Please spell out/explain abbreviations and symbols (ps, lambda and R). Authors: Modifications made.

Comment 21: Referee #2: Figure 6: Please spell out MAGST and ACR and explain the meaning of numbered locations (also Fig. 8).

Authors: Ok, sensors are represented as points and the number are their labels (ID).

Comment 22: Referee #2: Figure 7: Please spell out MAGST. How was the PISR variable computed (not mentioned in the methods section)?

Authors: We agree. We added a description in the methods section to explain how the PISR values were obtained (lines 207-209)

Please also note the supplement to this comment:
http://www.the-cryosphere-discuss.net/tc-2016-211/tc-2016-211-AC2-supplement.zip

---

## Author Response (AR2)

**Author's response to the editor comments**

Please find our responses to the comments addressed by the editor for the manuscript tc-2016-211 followed by the marked-up manuscript version. The new versions of the files (after correction) have been uploaded under the names "tc-2016-211_Abstract_final", "tc-2016-211_manuscript_final" and "tc-2016-211_supplement_final"

5  **Editor comments**

**P1, Abstract: I miss some more information (a sentence or two) in the abstract and conclusions related to your results on the snow physical and thermal properties. You made great efforts on snow thickness sounding, excavation of snow pits for observations of snow stratigraphy and measurement of density and temperature variations – which were important contributions to your study**.

Response:

We added a sentence in the abstract (lines 19 to 23) and in the conclusion ( line 476) to specify the snow properties

**P2, L60-63: The drill sites presented in Isaksen et al. (2001, 2007) have deep and extensive permafrost with MAGT at**
15  **e.g. 100 m depth between -2.5 and 3 °C. Studies from Scandinavia at similar sites as yours that address the influence of snow cover on mountain permafrost at wind-swept sites are published in Farbrot et al., (2011, 2013) and Isaksen et al. 2011. Thus you should replace Isaksen et al. (2001, 2007) with these references.**

20  Response:

We agree, we added Farbrot et al (2011; 2013) (line 62) and removed Isaksen et al (2001, 2007)

**P17, L510: In other parts of the paper 30 cm is used. Try to be consistent on these values or use e.g. 30-40 cm.**

Response:

25  We agree, modification made.

Some very minor additional changes have been made by the authors (e.g. typos correction; turns-of-phrase). All these changes appear in the marked-up manuscript version.

[revised manuscript text omitted]